# ⊡ EigenGame Unloaded ⊡
# When playing games is better than optimizing

**Ian Gemp**[*], **Brian McWilliams**[*], **Claire Vernade & Thore Graepel**
DeepMind, London UK
`{imgemp,bmcw,vernade}@deepmind.com, thoregraepel@gmail.com`

## Abstract

We build on the recently proposed EigenGame that views eigendecomposition as a competitive game. EigenGame's updates are biased if computed using minibatches of data, which hinders convergence and more sophisticated parallelism in the stochastic setting. In this work, we propose an unbiased stochastic update that is asymptotically equivalent to EigenGame, enjoys greater parallelism allowing computation on datasets of larger sample sizes, and outperforms EigenGame in experiments. We present applications to finding the principal components of massive datasets and performing spectral clustering of graphs. We analyze and discuss our proposed update in the context of EigenGame and the shift in perspective from optimization to games.

## 1 Introduction

Large, high-dimensional datasets containing billions of samples are commonplace. Dimensionality reduction to extract the most informative features is an important step in the data processing pipeline which enables faster learning of classifiers and regressors (Dhillon *et al.*, 2013), clustering (Kannan and Vempala, 2009), and interpretable visualizations. Many dimensionality reduction and clustering techniques rely on eigendecomposition at their core including principal component analysis (Jolliffe, 2002), locally linear embedding (Roweis and Saul, 2000), multidimensional scaling (Mead, 1992), Isomap (Tenenbaum *et al.*, 2000), and graph spectral clustering (Von Luxburg, 2007).

Numerical solutions to the eigenvalue problem have been approached from a variety of angles for centuries: Jacobi's method, Rayleigh quotient, power (von Mises) iteration (Golub and Van der Vorst, 2000). For large datasets that do not fit in memory, approaches that access only subsets—or *minibatches*—of the data at a time have been proposed.

Recently, EigenGame (Gemp *et al.*, 2021) was introduced with the novel perspective of viewing the set of eigenvectors as the Nash strategy of a suitably defined game. While this work demonstrated an algorithm that was empirically competitive given access to only subsets of the data, its performance degraded with smaller minibatch sizes, which are required to fit high dimensional data onto devices.

One path towards circumventing EigenGame's need for large minibatch sizes is parallelization. In a data parallel approach, updates are computed in parallel on partitions of the data and then combined such that the aggregate update is equivalent to a single large-batch update. The technical obstacle preventing such an approach for EigenGame lies in the bias of its updates, i.e., the divide-and-conquer EigenGame update is not equivalent to the large-batch update. Biased updates are not just a theoretical nuisance; they can slow and even prevent convergence to the solution (made obvious in Figure 4).

In this work we introduce a formulation of EigenGame which admits unbiased updates which we term $\mu$-EigenGame. We will refer to the original formulation of EigenGame as $\alpha$-EigenGame.[1]

$\mu$-EigenGame and $\alpha$-EigenGame are contrasted in Figure 1. Unbiased updates allow us to increase the effective batch size using data parallelism. Lower variance updates mean that $\mu$-EigenGame should converge faster and to more accurate solutions than $\alpha$-EigenGame regardless of batch size. In Figure 1a (top), the density of the shaded region shows the distribution of steps taken by the

---

[*]denotes equal contribution.
[1]$\mu$ signifies unbiased or *un*loaded and $\alpha$ denotes original.

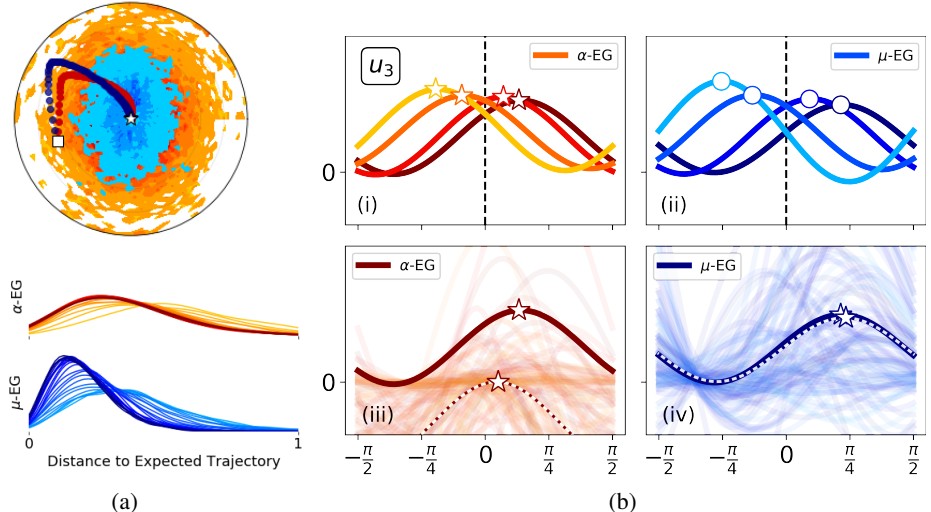

Figure 1: **(a)** Comparing $\alpha$-EigenGame (Gemp *et al.*, 2021) and $\mu$-EigenGame (this work) over 1000 trials with a batch size of 1. **(top)** The expected trajectory[2] of each algorithm from initialization ($\square$) to the true value of the third eigenvector ($\star$). **(bottom)** The distribution of distances between stochastic update trajectories and the expected trajectory of each algorithm as a function of iteration count (**bolder** lines are later iterations and modes further left are more desirable).

**(b)** Empirical support for Lemma 2. In the top row, player 3's utility is given for parents mis-specified by an angular distance along the sphere of $\angle(\hat{v}_{j<i}, v_{j<i}) \in [-20°, -10°, 10°, 20°]$ moving from light to dark. Player 3's mis-specification, $\angle(\hat{v}_i, v_i)$, is given by the x-axis (optimum is at 0 radians). $\alpha$-EigenGame (i) exhibits slightly lower sensitivity than $\mu$-EigenGame (ii) to mis-specified parents (see equation (8)). However, when the utilities are estimated using samples $X_t \sim p(X)$ (faint lines), $\mu$-EigenGame remains accurate (iv), while $\alpha$-EigenGame (iii) returns a utility (dotted line) with an optimum that is shifted to the left and down. The downward shift occurs because of the random variable in the denominator of the penalty terms (see equation (3)).[3]

---

[2]The trajectory when updating with $\mathbb{E}[X_t^\top X_t]$.

[3]Overestimation is expected by Jensen's: $\mathbb{E}[\frac{1}{X}] \geq \frac{1}{\mathbb{E}[X]}$.

stochastic variant of each algorithm after 100 burn-in steps. Although the expected path of $\alpha$-EG is slightly more direct, its stochastic variant has much larger variance. Figure 1a (bottom) shows that with increasing iterations, the $\mu$-EG trajectory approaches its expected value whereas $\alpha$-EG exhibits larger bias. Figure 1b further supports $\mu$-EigenGame's reduced bias with details in Sections 3 and 4.

**Our contributions**: In the rest of the paper, we present our new formulation of EigenGame, analyze its bias and propose a **novel unbiased parallel variant, $\mu$-EigenGame** with *stochastic* convergence guarantees. $\mu$-EigenGame's utilities are distinct from $\alpha$-EigenGame and offer an alternative perspective. We demonstrate its performance with extensive experiments including dimensionality reduction of massive data sets and clustering a large social network graph. We conclude with discussions of the algorithm's design and context within optimization, game theory, and neuroscience.

## 2 PRELIMINARIES AND RELATED WORK

In this work, we aim to compute the top-$k$ right singular vectors of data $X$, which is either represented as a matrix, $X \in \mathbb{R}^{n \times d}$, of $n$ $d$-dimensional samples, or as a $d$-dimensional random variable. In either case, we assume we can repeatedly sample a minibatch $X_t$ from the data of size $n' < n$, $X_t \in \mathbb{R}^{n' \times d}$. The top-$k$ right singular vectors of the dataset are then given by the top-$k$ eigenvectors of the (sample) covariance matrix, $C = \mathbb{E}[\frac{1}{n'} X_t^\top X_t] = \mathbb{E}[C_t]$.

For small datasets, SVD is appropriate. However, the time, $\mathcal{O}(\min\{nd^2, n^2d\})$, and space, $\mathcal{O}(nd)$, complexity of SVD prohibit its use for larger datasets (Shamir, 2015) including when $X$ is a random variable. For larger datasets, stochastic, randomized, or sketching algorithms are better suited. Stochastic algorithms such as Oja's algorithm (Oja, 1982; Allen-Zhu and Li, 2017) perform power

iteration (Rutishauser, 1971) to iteratively improve an approximation, maintaining orthogonality of the eigenvectors typically through repeated QR decompositions. Alternatively, randomized algorithms (Halko *et al.*, 2011; Sarlos, 2006; Cohen *et al.*, 2017) first compute a random projection of the data onto a $(k + p)$-subspace approximately containing the top-$k$ subspace. This is done using techniques similar to Krylov subspace iteration methods (Musco and Musco, 2015). After projecting, a call to SVD is then made on this reduced-dimensionality data matrix. Sketching algorithms (Feldman *et al.*, 2020) such as Frequent Directions (Ghashami *et al.*, 2016) also target learning the top-$k$ subspace by maintaining an overcomplete sketch matrix of size $(k + p) \times d$ and maintaining a span of the top subspace with repeated calls to SVD. In both the randomized and sketching approaches, a final SVD of the $n \times (k + p)$ dataset is required to recover the desired singular vectors. Although the SVD scales linearly in $n$, some datasets are too large to fit in memory; in this case, an out-of-memory SVD may suffice (Haidar *et al.*, 2017). For this reason, the direct approach of stochastic algorithms, which avoid an SVD call altogether, is appealing when processing very large datasets.

A large literature on *distributed* approaches to PCA exists (Liang *et al.*, 2014; Garber *et al.*, 2017; Fan *et al.*, 2019). These typically follow the pattern of computing solutions locally and then aggregating them in a single round (or minimal rounds) of communication. The modern distributed machine learning setting which has evolved to meet the needs of deep learning is fundamentally different. Many accelerators joined with fast interconnects means the cost of communication is low compared to the cost of a single update step, however existing approaches to distributed PCA cannot take full advantage of this.

**Notation**: We follow the same notation as Gemp *et al.* (2021). Variables returned by an approximation algorithm are distinguished from the true solutions with hats, e.g., the column-wise matrix of eigenvectors $\hat{V}$ approximates $V$. We order the columns of $V$ such that the $i$th column, $v_i$, is the eigenvector with the $i$th largest eigenvalue

---

**Algorithm 1** $\mu$-EigenGame$^R$

1: Given: data stream $X_t \in \mathbb{R}^{n' \times d}$, vectors $\hat{v}_i^0 \in \mathcal{S}^{d-1}$, step sequence $\eta_t$, and iterations $T$.
2: $\hat{v}_i \leftarrow \hat{v}_i^0$ for all $i$
3: **for** $t = 1 : T$ **do**
4:     **parfor** $i = 1 : k$ **do**
5:         `rewards` $\leftarrow \frac{1}{n'} X_t^\top X_t \hat{v}_i$
6:         `penalties` $\leftarrow$    $\frac{1}{n'} \sum_{j<i} \langle X_t \hat{v}_i, X_t \hat{v}_j \rangle \hat{v}_j$
7:         $\tilde{\nabla}_i^\mu \leftarrow$ `rewards` $-$ `penalties`
8:         $\tilde{\nabla}_i^{\mu,R} \leftarrow \tilde{\nabla}_i^\mu - \langle \tilde{\nabla}_i^\mu, \hat{v}_i \rangle \hat{v}_i$
9:         $\hat{v}_i' \leftarrow \hat{v}_i + \eta_t \tilde{\nabla}_i^{\mu,R}$
10:       $\hat{v}_i \leftarrow \frac{\hat{v}_i'}{||\hat{v}_i'||}$
11:     **end parfor**
12: **end for**
13: return all $\hat{v}_i$

---

$\lambda_i$. The set of all eigenvectors $\{v_j\}$ with $\lambda_j$ larger than $\lambda_i$, namely $v_i$'s *parents*, will be denoted by $v_{j<i}$. Similarly, sums over subsets of indices may be abbreviated as $\sum_{j<i} = \sum_{j=1}^{i-1}$. The set of all parents *and* children of $v_i$ are denoted by $v_{-i}$. Let the $i$th eigengap $g_i = \lambda_i - \lambda_{i+1}$. We assume the standard Euclidean inner product $\langle u, v \rangle = u^\top v$ and denote the unit-sphere and simplex in ambient space $\mathbb{R}^d$ with $\mathcal{S}^{d-1}$ and $\Delta^{d-1}$ respectively.

$\alpha$**-EigenGame.** We build on the algorithm introduced by Gemp *et al.* (2021), which we refer to here as $\alpha$-EigenGame. This algorithm is derived by formulating the eigendecomposition of a symmetric positive definite matrix as the Nash equilibrium of a game among $k$ players, each player $i$ owning the approximate eigenvector $\hat{v}_i \in \mathcal{S}^{d-1}$. Each player is also assigned a utility function, $u_i^\alpha(\hat{v}_i | \hat{v}_{j<i})$, that they must maximize:

$$u_i^\alpha(\hat{v}_i | \hat{v}_{j<i}) = \overbrace{\hat{v}_i^\top C \hat{v}_i}^{\text{Var}} - \sum_{j<i} \overbrace{\frac{\langle \hat{v}_i, C\hat{v}_j \rangle^2}{\langle \hat{v}_j, C\hat{v}_j \rangle}}^{\text{Align-penalty}} . \tag{1}$$

These utilities balance two terms, one that rewards a $\hat{v}_i$ that captures more variance in the data and a second term that penalizes $\hat{v}_i$ for failing to be orthogonal to each of its parents $\hat{v}_{j<i}$ (these terms are indicated with Var and Align-penalty in equation (1)). In $\alpha$-EigenGame, each player simultaneously updates $\hat{v}_i$ with gradient ascent, and it is shown that this process converges to the Nash equilibrium. We are interested in extending this approach to the data parallel setting where each player $i$ may distribute its update computation over multiple devices.

## 3 A SCALABLE UNBIASED ALGORITHM

We present our novel modification to $\alpha$-EigenGame called $\mu$-EigenGame along with intuition, theory, and empirical support for critical lemmas. We begin with identifying and systematically removing the bias that exists in the $\alpha$-EigenGame updates. We then explain how removing bias allows us to exploit modern compute architectures culminating in the development of a highly parallelizable algorithm.

### 3.1 $\alpha$-EIGENGAME'S BIASED UPDATES

Consider partitioning the sample covariance matrix $C_t$ into a sum of $m$ matrices as $C_t = \frac{1}{n'} X_t^\top X_t = \frac{1}{m} \sum_m \frac{m}{n'} X_{tm}^\top X_{tm} = \frac{1}{m} \sum_m C_{tm}$. For sake of exposition, we drop the additional subscript $t$ on $C$ in what follows. We would like $\alpha$-EigenGame to parallelize over these partitions. However, the gradient of $u_i^\alpha$ with respect to $\hat{v}_i$ does not decompose cleanly over the data partitions:

$$\nabla_i^\alpha \propto \overbrace{C\hat{v}_i}^{\texttt{Var}} - \sum_{j<i} \overbrace{\frac{\hat{v}_i^\top C\hat{v}_j}{v_j^\top C\hat{v}_j}}^{\texttt{Align-penalty}} C\hat{v}_j = \frac{1}{m} \sum_m \Big[ C_m\hat{v}_i - \sum_{j<i} \boxed{\frac{\hat{v}_i^\top C\hat{v}_j}{\hat{v}_j^\top C\hat{v}_j}} C_m\hat{v}_j \Big]. \tag{2}$$

We include the superscript $\alpha$ on the EigenGame gradient to differentiate it from the $\mu$-EigenGame direction later. The nonlinear appearance of $C$ in the penalty terms makes obtaining an unbiased gradient difficult. The quadratic term in the numerator of equation (2) could be made unbiased by using two sample estimates of $C$, one for each term. But the appearance of the term in the denominator does not have an easy solution. $C_m$ is likely singular for small $n'$ ($n' < d$) which increases the likelihood of a small denominator, i.e., a large penalty coefficient (boxed), if we were to estimate the denominator with samples. The result is an update that emphasizes penalizing orthogonality over capturing data variance. Techniques exist to reduce the bias of samples of ratios of random variables, but to our knowledge, techniques to obtain unbiased estimates are not available. This was conjectured by Gemp *et al.* (2021) as the reason for why $\alpha$-EigenGame performed worse with small minibatches.

### 3.2 REMOVING $\alpha$-EIGENGAME'S BIAS

It is helpful to rearrange equation (2) to shift perspective from estimating a penalty coefficient (in red) to estimating a penalty direction (in blue):

$$\nabla_i^\alpha \propto \frac{1}{m} \sum_m \Big[ C_m\hat{v}_i - \sum_{j<i} \hat{v}_i^\top C_m\hat{v}_j \frac{C\hat{v}_j}{\hat{v}_j^\top C\hat{v}_j} \Big]. \tag{3}$$

The penalty direction in equation (3) is still difficult to estimate. However, consider the case where $\hat{v}_j$ is any eigenvector of $C$ with associated (unknown) eigenvalue $\lambda'$. In this case, $C\hat{v}_j = \lambda' \hat{v}_j$ and the penalty direction (in blue) simplifies to $\hat{v}_j$ because $||\hat{v}_j|| = 1$. While this assumption is certainly not met at initialization, $\alpha$-EigenGame leads each $\hat{v}_j$ towards $v_j$, so we can expect this assumption to be met asymptotically.

This intuition motivates the following $\mu$-EigenGame update direction for $\hat{v}_i$ with inexact parents $\hat{v}_j$ (compare orange in equation (4) to blue in equation (3)):

$$\Delta_i^\mu = C\hat{v}_i - \sum_{j<i} (\hat{v}_i^\top C\hat{v}_j)\hat{v}_j = \frac{1}{m} \sum_m \Big[ C_m\hat{v}_i - \sum_{j<i} (\hat{v}_i^\top C_m\hat{v}_j)\hat{v}_j \Big]. \tag{4}$$

We use $\Delta$ instead of $\nabla$ because the direction is not a gradient (discussed later). Notice how the strictly linear appearance of $C$ in $\mu$-EigenGame allows the update to easily decompose over the data partitions in equation (4). The $\mu$-EigenGame update satisfies two important properties.

**Lemma 1** (Asymptotic equivalence). *The $\mu$-EigenGame direction, $\Delta_i^\mu$, with exact parents ($\hat{v}_j = v_j \ \forall \ j < i$) is equivalent to $\alpha$-EigenGame.*

*Proof.* We start with $\alpha$-EigenGame and add a superscript $e$ to its gradient to emphasize this is the gradient computed with exact parents ($\hat{v}_j = v_j$). Then simplifying, we find

$$\nabla_i^{\alpha,e} \propto C\hat{v}_i - \sum_{j<i} \frac{\hat{v}_i^\top Cv_j}{v_j^\top Cv_j} Cv_j = C\hat{v}_i - \sum_{j<i} \frac{\hat{v}_i^\top Cv_j}{v_j^\top \cancel{\lambda_j'} v_j} \cancel{\lambda_j'} v_j = C\hat{v}_i - \sum_{j<i} (\hat{v}_i^\top Cv_j)v_j = \Delta_i^\mu. \tag{5}$$

Therefore, once the first $(i-1)$ eigenvectors are learned, learning the $i$th eigenvector with $\mu$-EigenGame is equivalent to learning with $\alpha$-EigenGame. $\square$

**Lemma 2** (Zero bias). *Unbiased estimates of $\Delta_i^\mu$ can be obtained with samples from $p(X)$.*

*Proof.* Let $X \sim p(X)$ where $X \in \mathbb{R}^d$ and $p(X)$ is the uniform distribution over the dataset. Then

$$\mathbb{E}[\Delta_i^\mu] = \mathbb{E}[XX^\top]\hat{v}_i - \sum_{j<i}(\hat{v}_i^\top \mathbb{E}[XX^\top]\hat{v}_j)\hat{v}_j = C\hat{v}_i - \sum_{j<i}(\hat{v}_i^\top C\hat{v}_j)\hat{v}_j. \tag{6}$$

where all expectations are with respect to $p(X)$. $\square$

These two lemmas provide the foundation for a performant algorithm. The first enables convergence to the desired solution, while the second facilitates scaling to larger datasets. Algorithm 1 presents pseudocode for $\mu$-EigenGame where computation is parallelized over the $k$ players.

### 3.3 MODEL AND DATA PARALLELISM

In our setting we have a number of connected devices. Specifically we consider the parallel framework specified by TPUv3 available in Google Cloud, however our setup is applicable to any multi-host, multi-device system. The $\alpha$-EigenGame formulation (Gemp *et al.*, 2021) considers an extreme form of model parallelism (Figure 2a) where each device has its own unique set of eigenvectors.

In this work we further consider a different form of model and data parallelism which is directly enabled by having unbiased updates (Figure 2b). This enables $\mu$-EigenGame to deal with both high-dimensional problems as well as massive sample sizes. Here each set of eigenvectors is copied on $M$ devices. Update directions are computed on each device individually using a different data stream and then combined by summing or averaging. Updates are applied to a single copy and this is duplicated across the $M-1$ remaining devices. In this way, updates are computed using an $M\times$ larger effective batch size while still allowing device-wise model parallelism. This setting is particularly useful when the number of samples is very large. This form of parallelism is not possible using the original EigenGame formulation since it relies on combining *unbiased* updates. In this sense, the parallelism discussed in this work generalizes that introduced by Gemp *et al.* (2021).

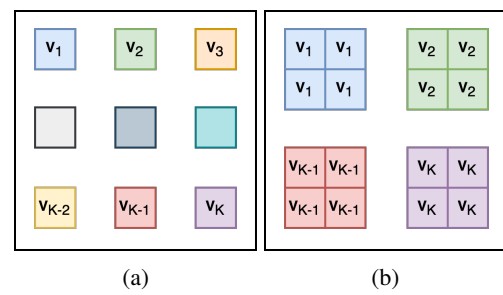

(a)     (b)

Figure 2: (a) Extreme model parallelism as proposed in $\alpha$-EigenGame. (b) Model and data parallelism enabled by $\mu$-EigenGame. Squares are separate devices (here, $M = 4$). Copies of estimates are color-coded. Updates are averaged across copies for a larger effective batch size.

Note that we also allow for within-device parallelism. That is, each $v_i$ in Figure 2 is a contiguous collection of eigenvectors which are updated independently, in parallel, on a given device (for example using `vmap` in Jax). We provide pseudocode in Algorithm 2 in the appendix which simply augments Algorithm 1 with an additional parallelized for-loop and aggregation step over available devices. We also provide detailed Jax pseudo-code for parallel $\mu$-EigenGame in Appendix F. We compare the empirical scaling performance of $\mu$-EigenGame against $\alpha$-EigenGame on a 14 billion sample dataset in section 5.

## 4 SVD AS THE SOLUTION TO A NEW EIGENGAME

We theoretically examine the $\mu$-EigenGame algorithm and 1) prove that, using only minibatches of data, $\mu$-EigenGame converges globally to the true eigenvectors, which 2) comprise the Nash equilibrium of a novel game formulation we recover through deriving *pseudo*-utility functions from update rules. Beyond proving specific theoretical properties of $\mu$-EigenGame, we believe these proof techniques may be of wider interest to the community.

### 4.1 CONVERGENCE TO SVD

The asymptotic equivalence of $\mu$-EigenGame to $\alpha$-EigenGame ensures $\mu$-EigenGame is globally, asymptotically convergent and its unbiased updates ensure it is scalable. Proof in appendix C.

**Theorem 1** (Global convergence). *Given a positive definite covariance matrix $C$ with the top-$k$ eigengaps positive and a square-summable, not summable step size sequence $\eta_t$ (e.g., $1/t$), Algorithm 1 converges to the top-$k$ eigenvectors asymptotically ($\lim_{T \to \infty}$) with probability $1$.*

This stochastic asymptotic convergence result is complimentary to the deterministic (full-batch) finite-sample result in Gemp *et al.* (2021) where each $\hat{v}_i$ is learned in sequence. In contrast, the proof above applies when learning all $\hat{v}_i$ in parallel. We leave finite-sample convergence to future work (Durmus *et al.*, 2020).

### 4.2 SVD IS NASH OF $\mu$-EIGENGAME

We arrived at $\mu$-EigenGame by analyzing and improving properties of the $\alpha$-EigenGame update. However, the $\mu$-EigenGame update direction is linear in each $\hat{v}_i$. This suggests we may be able to design a *pseudo*-utility function for it. Rearranging the update direction from equation (4) as

$$\Delta_i^\mu = C\hat{v}_i - \sum_{j<i} \hat{v}_j(\hat{v}_j^\top C\hat{v}_i) = \Big[I - \sum_{j<i} \hat{v}_j\hat{v}_j^\top\Big]C\hat{v}_i = \tilde{\nabla}_i^\mu \tag{7}$$

reveals that we can reverse-engineer the following utility function

$$u_i^\mu = \hat{v}_i^\top \Big[\overbrace{I - \sum_{j<i} \hat{v}_j\hat{v}_j^\top}^{\text{deflation}}\Big]C \bullet [\hat{v}_i] \tag{8}$$

where $\bullet$ is the stop gradient operator commonly used in deep learning packages. As the name implies, $\bullet$ stops gradients from flowing through its argument so that equation (8) appears linear in $\hat{v}_i$ instead of quadratic when differentiating the expression. In light of this, we have renamed $\Delta_i^\mu$ to $\tilde{\nabla}_i^\mu$ to emphasize that it is a *pseudo*-gradient of $u_i^\mu$. Note that without the stop gradient, the true gradient of $u_i^\mu$ would be $[A + A^\top]\hat{v}_i$ rather than $A\hat{v}_i$ where $A = [I - \sum_{j<i} \hat{v}_j^\top \hat{v}_j]C$. We analyze this alternative in Appendix H.1 and find it, interestingly, to perform worse than $\mu$-EigenGame empirically.

The utility function $u_i^\mu$ has an intuitive meaning. It is the Rayleigh quotient for the matrix $C_i = [I - \sum_{j<i} \hat{v}^\top \hat{v}_j]C$, which gives the covariance after the subspace spanned by $\hat{v}_{j<i}$ has been removed. In other words, player $i$ is directed to find the largest eigenvalue in the orthogonal complement of the approximate top-$(i-1)$ subspace. This approach is known as "deflating" the matrix $C$. Figure 1b illustrates $\mu$-EigenGame's reduced bias when estimating the new utility function (and resulting optimum) from an average over minibatches.

**Definition 1** ($\mu$-Eigen**Game**). *Let $\mu$-EigenGame be the game with players $i \in \{1, \ldots, k\}$, their respective strategy spaces $\hat{v}_i \in \mathcal{S}^{d-1}$, and their corresponding utilities $u_i^\mu$ as defined in equation (8).*

**Theorem 2.** *Top-$k$ SVD is the unique Nash of $\mu$-EigenGame given symmetric $C$ with the top-$k$ eigengaps positive.*

*Proof.* We will show by induction that each $v_i$ is the unique best response to $v_{-i}$, which implies they constitute the unique Nash equilibrium. First, consider player 1's utility. It is the Rayleigh quotient of $C$ because $\hat{v}_1$ is constrained to the unit-sphere, i.e., $u_1^\mu = \hat{v}_1^\top C\hat{v}_1 = \frac{\hat{v}_1^\top C\hat{v}_1}{\hat{v}_1^\top \hat{v}_1}$. Therefore, we know $v_1$ maximizes $u_1^\mu$ and the maximizer is unique because its eigengap $g_1 > 0$. In game theory parlance, $v_1$ is a *best response* to $v_{-1}$. The proof continues by induction. The utility of player $i$ is $u_i^\mu = \hat{v}_i^\top[I - \sum_{j<i} v_j v_j^\top]C\hat{v}_i$, which is the Rayleigh quotient with the subspace spanned by the top $(i-1)$ eigenvectors removed. Therefore, the maximizer of $u_i^\mu$ is the largest eigenvector in the remaining subspace, i.e., $v_i$. As before, $g_i > 0$, so this maximizer is unique. This shows that each $v_i$ is the unique best response to $v_{-i}$, therefore, the set of $v_i$ forms the unique Nash. $\square$

Notice how the induction proof of Theorem 2 relies on a) the hierarchy of vectors ($v_1$ does not depend on $v_{-1}$) and b) the fact that $u_i^\mu$ need only be a sensible utility when all player $i$'s parents

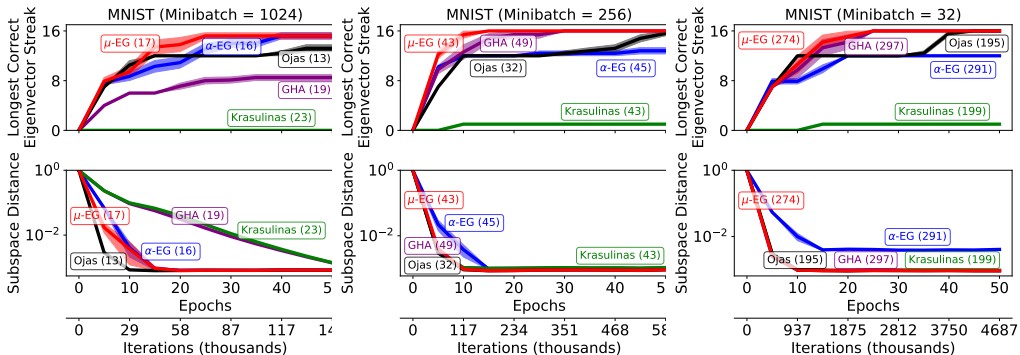

Figure 3: MNIST Experiment. Runtime (seconds) in legend on CPU ($m = 1$). Each column evaluates a different minibatch size $\in \{1024, 256, 32\}$. Shading indicates $\pm$ standard error of the mean. Learning rates were chosen from $\{10^{-3}, \ldots, 10^{-6}\}$ on 10 held out runs. Solid lines denote results with the best performing learning rate. All plots show means over 10 trials (randomness arising from minibatches and initialization). Shaded regions highlight $\pm$ standard error of the mean.

are eigenvectors. We revisit this in conjunction with Figure 5b later in discussion section 6.1 to aid researchers in the design of future approaches.

The Nash property is important because it enables the use of any black-box procedure for computing best responses. Like prior work, we develop a gradient method for optimizing each utility, however, that is not a requirement. Any approach suffices if it can efficiently compute a best response.

## 5 EXPERIMENTS

As in EigenGame, we omit the projection of gradients onto the tangent space of the sphere; specifically, we omit line 8 in Algorithm 1. As discussed in Gemp *et al.* (2021), this has the effect of intelligently adapting the step size to use smaller learning rates near the fixed point. To ease comparison with previous work, we count the *longest correct eigenvector streak* as introduced by Gemp *et al.* (2021), which measures the number of eigenvectors that have been learned, in order, to within an angular threshold (e.g., $\pi/8$) of the true eigenvectors. We also measure how well the set of $\hat{v}_i$ captures the top-$k$ subspace with a normalized subspace distance: $1 - 1/k \cdot \text{Tr}(U^*P) \in [0, 1]$ where $U^* = VV^\dagger$ and $P = \hat{V}\hat{V}^\dagger$ (Tang, 2019). We provide additional experiments in Appendix A.

**MNIST.** We compare $\mu$-EigenGame against $\alpha$-EigenGame, GHA (Sanger, 1989), Matrix Krasulina (Tang, 2019), and Oja's algorithm (Allen-Zhu and Li, 2017) on the MNIST dataset. We flatten each image in the training set to obtain a $60,000 \times 784$ dimensional matrix $X$. Figure 3 demonstrates $\mu$-EigenGame's robustness to minibatch size. It performs best in the longest streak metric and better than $\alpha$-EigenGame in subspace distance. We attribute this improvement to its unbiased updates and additional acceleration effects which we discuss in detail in section H.2.

**Meena conversational model.** This dataset consists a subset of the 40 billion words used to train the transformer-based Meena language model (Adiwardana *et al.*, 2020). The subset was preprocessed to remove duplicates and then embedded using the trained model.

The dataset consists of $n \approx 14$ billion embeddings each with dimensionality $d = 2560$; its total size is 131TB. Due to its moderate dimensionality we can exactly compute the ground truth solution by iteratively accumulating the covariance matrix of the data and computing its eigendecomposition. On a single machine this takes 1.5 days (but is embarrassingly parallelizable with MapReduce).

We use minibatches of size 4,096 in each TPU. We do model parallelism across 4 TPUs so we see 16,384 samples per iteration. We test two additional degrees of data parallelism with $4\times$ (16 TPUs, 65,536 samples) and $8\times$ (32 TPUs, 131,072 samples) the amount of data per iteration respectively. We compute and apply updates using SGD with a learning rate of $5 \times 10^{-5}$ and Nesterov momentum with a factor of 0.9.

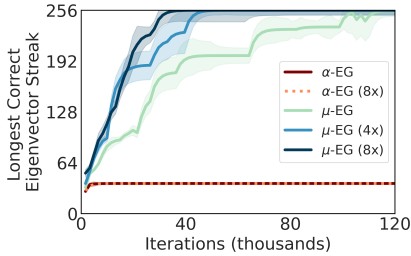

Figure 4: Comparison between $\mu$-EigenGame and $\alpha$-EigenGame with different degrees of data parallelism (in parentheses) on the Meena dataset.

Figure 4 compares the mean performance of $\mu$-EigenGame against $\alpha$-EigenGame as a function of the degree of parallelism in computing the top $k = 256$ eigenvectors (standard errors computed over 5 random seeds). Each TPU is tasked with learning 32 contiguous eigenvectors. We see that increasing the degree of parallelism has no effect on the performance of $\alpha$-EigenGame. As expected, it is unable to take advantage of the higher data throughput since its updates are biased and cannot be meaningfully linearly combined across copies. In contrast, the performance of $\mu$-EigenGame scales with the effective batch size achieved through parallelism. $\mu$-EigenGame ($8\times$) is able to recover 256 eigenvectors in less than 40,000 iterations in 2 hours 45 minutes (approximately 0.5 epochs).

**Spectral clustering on graphs.** We conducted an experiment on learning the eigenvectors of the graph Laplacian of a social network graph (Leskovec and McAuley, 2012) for the purpose of spectral clustering. The eigenvalues of the graph Laplacian reveal several interesting properties as well such as the number of connected components, an approximation to the sparsest cut, and the diameter of a connected graph (Chung *et al.*, 1994).

Given a graph with a set of nodes $\mathcal{V}$ and set of edges $\mathcal{E}$, the graph Laplacian can be written as $\mathcal{L} = X^\top X$ where each row of the incidence matrix $X \in \mathbb{R}^{|\mathcal{E}| \times |\mathcal{V}|}$ represents a distinct edge; $X_{e=(i,j)\in\mathcal{E}}$ is a vector containing only 2 nonzero entries, a $1$ at index $i$ and a $-1$ at index $j$ (Horaud, 2009). In this setting, the eigenvectors of primary interest are the bottom-$k$ ($\lambda_{|\mathcal{V}|}, \lambda_{|\mathcal{V}|-1}, \ldots$) rather than the top-$k$ ($\lambda_1, \lambda_2, \ldots$), however, a simple algebraic manipulation allows us to reuse a top-$k$ solver. By defining the matrix $\mathcal{L}^- = \lambda^* I - \mathcal{L}$ with $\lambda^* > \lambda_1$, we ensure $\mathcal{L}^- \succ 0$ and the top-$k$ eigenvectors of $\mathcal{L}^-$ are the bottom-$k$ of $\mathcal{L}$. The update in equation (4) is transformed into $\tilde{\nabla}_i^\mu = (\lambda^* I - \mathcal{L})\hat{v}_i - \sum_{j<i} \left( \hat{v}_i^\top (\lambda^* I - \mathcal{L})\hat{v}_j \right) \hat{v}_j$. We provide efficient pseudo-code in Appendix G.

The Facebook graph consists of $134,833$ nodes, $1,380,293$ edges, and $8$ connected components, each formed by a set of Facebook pages belonging to a distinct category, e.g., Government, TV shows, etc. (Leskovec and Krevl, 2014; Rozemberczki *et al.*, 2019). We add a single edge between every pair of components to create a connected graph. By projecting this graph onto the bottom $8$ eigenvectors of the graph Laplacian using $\mu$-EG ($M = 1, n' = \eta_t = \frac{|\mathcal{E}|}{1000}$) and then running $k$-means clustering (Pedregosa *et al.*, 2011), we are able to recover the ground truth clusters (see Figure 5a) with $99.92\%$ accuracy. The experiment was run on a single CPU.

## 6 DISCUSSION

### 6.1 UTILITIES TO UPDATES AND BACK

Figure 5b summarizes the relationships advising the designs of the various EigenGame algorithms. Starting from the $\alpha$-EigenGame utility, its update is arrived at by simply following the standard gradient ascent paradigm. In noticing that stochastic estimates of the gradient are biased, we arrive at the $\mu$-EigenGame update by considering how to remove this bias in a principled manner.

Sacrificing the exact steepest decent direction for a direction that allows unbiased estimates is a tradeoff that in this case has benefits. Also, while $\tilde{\nabla}_i^\mu$ is not a gradient (except with exact parents), the new penalties have properties (above) that make them intuitively more desirable than the originals; they are adaptive to the state of the system (discussed further in section H.2).

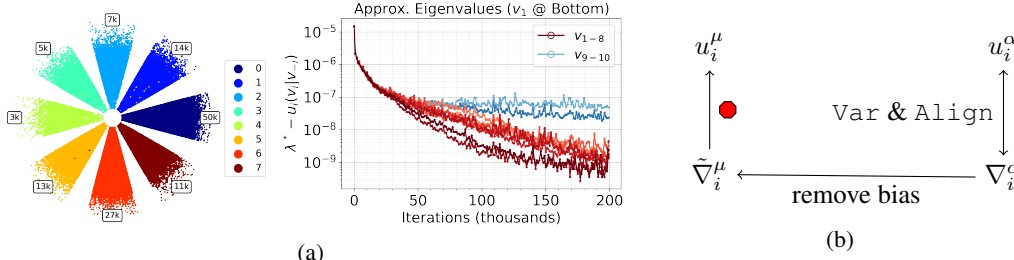

(a)          (b)

Figure 5: (5a) Facebook Page Networks. (Left) Petals differentiate ground truth clusters; colors differentiate learned clusters. Petals are ideally colored according to the color bar starting with the rightmost petal and proceeding counterclockwise. Numbers indicate ground truth cluster size. Clusters are extracted by running $k$-means clustering on the learned eigenvectors $\hat{V} \in \mathbb{R}^{|\mathcal{V}| \times k}$ (samples on rows). (Right) Rayleigh quotient plot reveals a gap between the 8th and 9th eigenvalues indicating $\approx 8$ clusters exist. (5b) Relationships between utilities and updates. An arrow indicates the endpoint is reasonably derived from the origin; the lack of an arrow indicates the direction is unlikely.

We derive pseudo-utilities with desired theoretical properties by *integrating* the new updates with help from the stop gradient operator. However, it is unlikely that this utility would be developed independently of these steps to solve the problem at hand (see Appendix H for more details). This suggests an alternative approach to algorithm design complementary to the optimization perspective: directly designing updates themselves which converge to the desired solution, reminiscent of previous paradigms that drove neuro-inspired learning rules.

## 6.2 BRIDGING HEBBIAN AND OPTIMIZATION APPROACHES

The Generalized Hebbian Algorithm (GHA) (Sanger, 1989; Gang *et al.*, 2019; Chen *et al.*, 2019) update direction for $\hat{v}_i$ with inexact parents $\hat{v}_j$ is similar to $\mu$-EigenGame:

$$\Delta_i^{gha} = C\hat{v}_i - \sum_{j \leq i} (\hat{v}_i^\top C \hat{v}_j)\hat{v}_j. \tag{9}$$

$C$ appears linearly in this update so GHA can also be parallelized. In contrast to $\mu$-EigenGame, GHA additionally penalizes the alignment of $\hat{v}_i$ to itself and removes the unit norm constraint on $\hat{v}_i$ (not shown). Without any constraints, GHA overflows in experiments. We take the approach of Gemp *et al.* (2021) and constrain $\hat{v}_i$ to the unit-ball ($||\hat{v}_i|| \leq 1$) rather than the unit-sphere ($||\hat{v}_i|| = 1$).

The connection between GHA and $\mu$-EigenGame is interesting because unlike $\mu$-EigenGame, GHA is a Hebbian learning algorithm inspired by neuroscience and its update rule is not motivated from the perspective of maximizing of a utility function. Game formulations of classical machine learning problems may provide a bridge between statistical and biologically inspired viewpoints.

## 7 CONCLUSION

We introduced $\mu$-EigenGame, an unbiased, globally convergent, parallelizable algorithm that recovers the top-$k$ eigenvectors of a symmetric positive definite matrix. We demonstrated the performance of $\mu$-EigenGame on large scale dimension reduction and clustering problems. We discussed technical details of $\mu$-EigenGame within the context of game theory, machine learning and neuroscience.

Like its predecessor, $\mu$-EigenGame is a $k$-player, general-sum game allowing model parallelism over players; our unbiased reformulation allows even greater parallelism over data. Furthermore, the hierarchy and Nash property enable the exploration of more sophisticated best responses.

$\mu$-EigenGame's improved robustness to smaller minibatches makes it more amenable to being used as part of deep learning, optimization (Krummenacher *et al.*, 2016), and regularization (Miyato *et al.*, 2018) techniques which leverage spectral information of gradient covariances or Hessians. Graph spectral methods have also recently shown to be related to state-of-the-art representation learning algorithms (HaoChen *et al.*, 2021) further cementing the importance of efficient SVD algorithms in modern machine learning.

**Acknowledgements.**    We would like to thank Trevor Cai, Rosalia Schneider, Dimitrios Vytiniotis for invaluable help with optimizing algorithm performance on TPU. We also thank Maribeth Rauh, Zonglin Li, Daniel Adiwardana and the Meena team for providing us with data and assistance. And finally, we thank Alexander Novikov for helpful feedback on the manuscript.

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

## A    EXPERIMENTS ON SYNTHETIC DATA

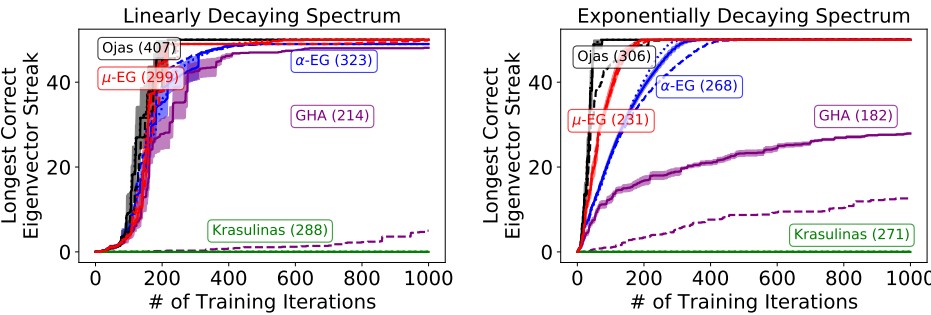

Figure 6: Synthetic Experiment. Runtime (milliseconds) in legend.

We validate $\mu$-EigenGame in a full-batch setting on two synthetic datasets: one with exponentially decaying spectrum; the other with a linearly decaying spectrum. Figure 6 shows $\mu$-EigenGame outperforms $\alpha$-EigenGame on the former and matches its performance on the latter. We discuss possible reasons for this gap in the discussion in Section 6.

## B    PARALLELIZED ALGORITHM

**Riemannian Manifolds.**    Before introducing an algorithm for $\mu$-EigenGame, we first briefly review necessary terminology for learning on Riemannian manifolds Absil *et al.* (2009), specifically for the sphere. The notation $\mathcal{T}_{\hat{v}_i} \mathcal{S}^{d-1}$ denotes the set of vectors tangent to the sphere at a point $\hat{v}_i$ (i.e., any vector orthogonal to $\hat{v}_i$). $R_{\hat{v}_i}(z) = \frac{\hat{v}_i + z}{||\hat{v}_i + z||}$ is the commonly used restriction of the retraction on $\mathcal{S}^{d-1}$ to the tangent bundle at $\hat{v}_i$ (i.e., step in tangent direction $z$ and then unit-normalize the result). The operator $\Pi_{\hat{v}_i}(y) = (I - \hat{v}_i^\top \hat{v}_i)y$ projects the direction $y$ onto $\mathcal{T}_{\hat{v}_i} \mathcal{S}^{d-1}$. Combining these tools together results in a movement along the Riemannian manifold: $\hat{v}_i^{(t+1)} \leftarrow R_{\hat{v}_i}\big(\Pi_{\hat{v}_i}(y)\big)$.

We present pseudocode for $\mu$-EigenGame below where computation is parallelized both over the $k$ players and over $M$ machines per player.

---

**Algorithm 2** $\mu$-EigenGame$^R$

---

1:  Given: data stream $X_t \in \mathbb{R}^{n' \times d}$, number of parallel machines $M$ per player (minibatch size per machine $n'' = \frac{n'}{M}$), initial vectors $\hat{v}_i^0 \in \mathcal{S}^{d-1}$, step size sequence $\eta_t$, and number of iterations $T$.
2:  $\hat{v}_i \leftarrow \hat{v}_i^0$ for all $i$
3:  **for** $t = 1 : T$ **do**
4:      **parfor** $i = 1 : k$ **do**
5:          **parfor** $m = 1 : M$ **do**
6:              rewards $\leftarrow X_{tm}^\top X_{tm} \hat{v}_i$
7:              penalties $\leftarrow \sum_{j<i} \langle X_{tm}\hat{v}_i, X_{tm}\hat{v}_j \rangle \hat{v}_j$
8:              $\tilde{\nabla}_{im}^\mu \leftarrow (\text{rewards} - \text{penalties})/n''$
9:              $\tilde{\nabla}_{im}^{\mu,R} \leftarrow \tilde{\nabla}_{im}^\mu - \langle \tilde{\nabla}_{im}^\mu, \hat{v}_i \rangle \hat{v}_i$
10:          **end parfor**
11:          $\tilde{\nabla}_i^{\mu,R} \leftarrow \frac{1}{M} \sum_m [\tilde{\nabla}_{im}^{\mu,R}]$
12:          $\hat{v}_i' \leftarrow \hat{v}_i + \eta_t \tilde{\nabla}_i^{\mu,R}$
13:          $\hat{v}_i \leftarrow \frac{\hat{v}_i'}{||\hat{v}_i'||}$
14:      **end parfor**
15:  **end for**
16:  return all $\hat{v}_i$

---

## C  GLOBAL STOCHASTIC CONVERGENCE

**Theorem 1** (Global Convergence). *Given a positive definite covariance matrix $C$ with the top-$k$ eigengaps positive and a square-summable, not summable step size sequence $\eta_t$ (e.g., $1/t$), Algorithm 1 converges to the top-$k$ eigenvectors asymptotically ($\lim_{T \to \infty}$) with probability $1$.*

*Proof.* Assume none of the $\hat{v}_i$ are initialized to an angle exactly $90°$ away from the true eigenvector: $\langle \hat{v}_i, v_i \rangle \neq 0$. The set of vectors $\{\hat{v}_i : \langle \hat{v}_i, v_i \rangle = 0\}$ has Lebesgue measure $0$, therefore, the above assumption holds w.p.1. The update direction for the top eigenvector $\hat{v}_1$ is exactly equal to that of $\alpha$-EigenGame ($\tilde{\nabla}_1^\mu = \nabla_1^\alpha$), therefore, they have the same limit points for $\hat{v}_1$. The proof then proceeds by induction. As $\hat{v}_{j<i}$ approach their limit points, the update for the $i$th eigenvector $\hat{v}_i$ approaches that of $\alpha$-EigenGame ($\tilde{\nabla}_i^\mu = \nabla_i^\alpha$) and, by Lemma 3, the stable region of $\mu$-EigenGame also shrinks to a point around the top-$k$ eigenvectors.

Denote the "update field" $H(v)$ to match the work of Shah (2019). $H(v)$ is simply the concatenation of all players' Riemannian update rules, i.e., all players updating in parallel using their Riemannian updates:

$$H(v) = [(I - \hat{v}_1 \hat{v}_1^\top)\Delta_1^\mu, \ldots, (I - \hat{v}_k \hat{v}_k^\top)\Delta_k^\mu] : \mathbb{R}^{kd} \to \mathbb{R}^{kd} \tag{10}$$

where $\Delta_i^\mu$ is defined in equation (7) and $(I - p_1 p_1^\top)\Delta_i^\mu$ projects $\Delta_i^\mu$ onto the tangent space of player $i$'s unit sphere.

The result is then obtained by applying Theorem 7 of Shah (2019) with the following information: a) the unit-sphere is a compact manifold with an injectivity radius of $\pi$, b) the update field is a polynomial in $\{v_i\}$ and therefore smooth (analytic), and c) by Lemma 4 (see Appendix E) the update noise constitutes a bounded martingale difference sequence. $\square$

While the convergence proof for $\alpha$-EG provides finite-sample rates, it only applies to the algorithm applied sequentially (not parallelized over eigenvectors) and in the deterministic setting (minibatch contains the entire data set). The experiments in Gemp *et al.* (2021) apply the algorithm in parallel and with mini batch sizes, meaning the $\alpha$-EG theorem does not actually apply to their experimental setting. That is to say, the $\alpha$-EG paper proposes updating eigenvectors in parallel in practice despite the lack of convergence guarantee.

In contrast, our convergence theorem applies to $\mu$-EG when applied in parallel (over the eigenvectors) and in the stochastic setting (with mini batch sizes), which is what we examine empirically in our experiments. The downside is that we do not provide finite-sample convergence rates.

Although we do not provide convergence rates, Lemma 1 proves that the $\mu$-EG update converges to the $\alpha$-EG update for each eigenvector, so intuitively, we expect the convergence rates to be relatively similar given that the algorithms are equivalent in the limit. Note that in the full batch setting where stochasticity does not conflate the differences between the two algorithms, Figure 6 in Appendix A empirically supports the similarity of the convergence rates for the two algorithms. Figure 3 (minibatch of $1024$) which looks at a large (but not full) minibatch size, also shows a small difference between convergence for the two algorithms.

In summary, the $\alpha$-EG convergence theorem is impractical—it provides convergence rates for a (non-parallel) algorithm in the (non-stochastic) setting, which is a combination that $\alpha$-EG paper does not suggest be applied in practice. In contrast, our $\mu$-EG convergence theorem is practical—it provides asymptotic convergence for a parallel algorithm in the stochastic setting.

**Difficulties Obtaining Finite Sample Rates**   In consideration of a finite sample convergence result, we consulted Durmus *et al.* (2020). The primary obstacle to applying their convergence theorem is the construction of a suitable Lyapunov function to satisfy their Assumption A.2 stated on page 4. Constructing Lyapunov functions is typically a tedious, unpredictable process. The work in Durmus *et al.* (2020) is very recent and finite sample convergence of Riemannian stochastic approximation (i.e., update directions are not gradients of any function) schemes is cutting edge, highly technical research. This is in contrast to Riemannian optimization (i.e., update directions are the gradient of a function), which is much more mature. We hope theory advances in the near future to a point where we can more easily provide convergence rates for algorithms like $\alpha$-EigenGame.

# D    ERROR PROPAGATION / SENSITIVITY ANALYSIS

**Lemma 3.** *An $\mathcal{O}(\epsilon)$ angular error of parent $\hat{v}_{j<i}$ implies an $\mathcal{O}(\epsilon)$ angular error in the location of the solution for $\hat{v}_i$.*

*Proof.* The proof proceeds in three steps:

1. $\mathcal{O}(\epsilon)$ angular error of parent $\implies \mathcal{O}(\epsilon)$ Euclidean error of parent

2. $\mathcal{O}(\epsilon)$ Euclidean error of parent $\implies \mathcal{O}(\epsilon)$ Euclidean error of norm of child gradient

3. $\mathcal{O}(\epsilon)$ Euclidean error of norm child gradient + instability of minimum at $\pm\frac{\pi}{2} \implies \mathcal{O}(\epsilon)$ angular error of child's solution.

Angular error in the parent can be converted to Euclidean error by considering the chord length between the mis-specified parent and the true parent direction. The two vectors plus the chord form an isoceles triangle with the relation that chord length $l = 2\sin(\frac{\epsilon}{2})$ is $\mathcal{O}(\epsilon)$ for $\epsilon \ll 1$.

Next, write the mis-specified parents as $\hat{v}_j = v_j + w_j$ where $||w_j||$ is $\mathcal{O}(\epsilon)$ as we have just shown. Let $b$ equal the difference between the Riemannian update direction $\tilde{\nabla}_i^\mu$ with approximate parents and that with exact parents. All directions we consider here are the Riemannian directions, i.e., they have been projected onto the tangent space of the sphere. Then

$$b = \tilde{\nabla}_i^{\mu,e} - \tilde{\nabla}_i^\mu = (\underbrace{I - \hat{v}_i\hat{v}_i^\top}_{\text{projection onto sphere}}) \sum_{j<i}\left[(\hat{v}_i^\top C\hat{v}_j)\hat{v}_j - (\hat{v}_i^\top Cv_j)v_j\right] \qquad (11)$$

and the norm of the difference is

$$\qquad (12)$$

$$||b|| = ||(I - \hat{v}_i\hat{v}_i^\top)\sum_{j<i}\left[(\hat{v}_i^\top C\hat{v}_j)\hat{v}_j - (\hat{v}_i^\top Cv_j)v_j\right]|| \qquad (13)$$

$$\leq ||I - \hat{v}_i\hat{v}_i^\top|| \cdot ||\sum_{j<i}\left[(\hat{v}_i^\top C\hat{v}_j)\hat{v}_j - (\hat{v}_i^\top Cv_j)v_j\right]|| \qquad (14)$$

$$\leq ||\sum_{j<i}\left[(\hat{v}_i^\top C\hat{v}_j)\hat{v}_j - (\hat{v}_i^\top Cv_j)v_j\right]||. \qquad (15)$$

We can further bound the summands with

$$||(\hat{v}_i^\top C\hat{v}_j)\hat{v}_j - (\hat{v}_i^\top Cv_j)v_j|| = ||(\hat{v}_j\hat{v}_j^\top - v_jv_j^\top)C\hat{v}_i|| \qquad (16)$$

$$\leq ||\hat{v}_j\hat{v}_j^\top - v_jv_j^\top||||C\hat{v}_i|| \qquad (17)$$

$$\leq \lambda_1||\hat{v}_j\hat{v}_j^\top - v_jv_j^\top|| \qquad (18)$$

$$= \lambda_1||(v_j + w_j)(v_j + w_j)^\top - v_jv_j^\top|| \qquad (19)$$

$$= \lambda_1||w_jv_j^\top + v_jw_j^\top + w_jw_j^\top|| \qquad (20)$$

$$\leq \lambda_1(||w_jv_j^\top|| + ||v_jw_j^\top|| + ||w_jw_j^\top||) \qquad (21)$$

$$= \mathcal{O}(\epsilon). \qquad (22)$$

This upper bound on the norm of the difference between the two directions translates to a lower bound on the inner product of the two directions wherever $||\tilde{\nabla}_i^{\mu,e}|| > \epsilon$, specifically $\langle \tilde{\nabla}_i^{\mu,e}, \tilde{\nabla}_i^\mu \rangle > 0$ (see Figure 7a). And recall that the direction with exact parents is equivalent to the gradient of $\alpha$-EigenGame with exact parents, $\nabla_i^{\alpha,e}$.

Therefore, by a Lyapunov argument, the $\tilde{\nabla}_i^\mu$ direction is an ascent direction on the $\alpha$-EigenGame utility where it forms an acute angle (positive inner product) with $\nabla_i^{\alpha,e}$. Furthermore, $\nabla_i^{\alpha,e}$ is the gradient of a utility function that is sinusoidal along the sphere manifold; specifically, it is a cosine with period $\pi$ and positive amplitude dependent on the spectrum of $C$ (c.f. equation (8) of Gemp

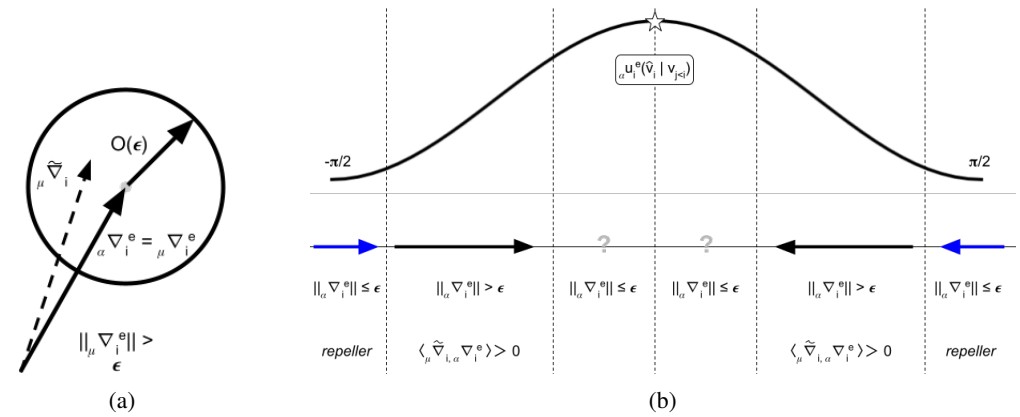

Figure 7: (a) Close in Euclidean distance can imply close in angular distance if the vectors are long enough. (b) The stable region for $\mu$-EigenGame consists of an $\mathcal{O}(\epsilon)$ ball around the true optimum as $\epsilon \to 0$.

*et al.* (2021)). We can derive an upper bound on the size of the angular region for which $\tilde{\nabla}_i^\mu$ is not necessarily an ascent direction (the "?" marks in Figure 7). This region is defined as the set of angles for which the norm of the utility's derivative is small, i.e., $||\nabla_i^{\alpha,e}|| \leq \epsilon$. The derivative of cosine is sine, which depends linearly on its argument (angle) for small values, therefore, $|\theta| \leq \mathcal{O}(\epsilon)$ or $|\frac{\pi}{2} - \theta| \leq \mathcal{O}(\epsilon)$. As long as $\hat{v}_i$ does not lie within the $|\frac{\pi}{2} - \mathcal{O}(\epsilon)|$ region, $\mu$-EigenGame will ascend the utility landscape to within $\mathcal{O}(\epsilon)$ angular error of the true eigenvector $v_i$. In the limit as $\epsilon \to 0$, the size of the $|\frac{\pi}{2} - \mathcal{O}(\epsilon)|$ region vanishes to a point, $v_i^\perp$. To understand the stability of this point, we can again appeal to the analysis from Gemp *et al.* (2021)—see equation (8) on page 7 of that work. The Jacobian of $\tilde{\nabla}_i^\mu$ and the Hessian of $u_i^\alpha$ are equal with exact parents, and we know that its Riemannian Hessian is positive definite if the $i$th eigengap is positive: $H_{\hat{v}_i}^R[u_i^\alpha] \succeq (\lambda_i - \lambda_{i+1})I$. This means that the point $v_i^\perp$ is a repeller for $\alpha$-EigenGame. Similarly to before, we can show more formally that an $\mathcal{O}(\epsilon)$ perturbation to parents results in an $\mathcal{O}(\epsilon)$ perturbation to the Jacobian of $\tilde{\nabla}_i^\mu$ from $H[u_i^\alpha]$:

$$J = [I - \sum_{j<i} \hat{v}_j \hat{v}_j^\top]C \tag{23}$$

$$= [I - \sum_{j<i} (v_j + w_j)(v_j + w_j)^\top]C \tag{24}$$

$$= [I - \sum_{j<i} v_j v_j^\top]C - \sum_{j<i} [w_j v_j^\top + v_j w_j^\top + w_j w_j^\top]C \tag{25}$$

$$= [I - \sum_{j<i} v_j v_j^\top]C - \mathcal{O}(\epsilon)W \tag{26}$$

$$= H[u_i^\alpha] - \mathcal{O}(\epsilon)W \tag{27}$$

where $W$ is some matrix with $\mathcal{O}(1)$ entries (w.r.t. $\epsilon$). For the sphere, the Riemannian Jacobian is a linear function of the Jacobian ($J_{\hat{v}_i}^R = (I - \hat{v}_i \hat{v}_i^\top)J - (\hat{v}_i^\top J \hat{v}_i)I = H_{\hat{v}_i}^R[u_i^\alpha] - \mathcal{O}(\epsilon)$) and therefore the error remains $\mathcal{O}(\epsilon)$. The set of (non)symmetric, positive semidefinite matrices ($A$ is p.s.d. iff $y^\top A y \geq 0 \; \forall \; y$) forms a closed convex cone, the interior of which contains positive definite matrices Wang *et al.* (2010). Therefore, $J_{\hat{v}_i}^R$ remains in this set after a small enough $\mathcal{O}(\epsilon)$ perturbation. Therefore, in the limit $\epsilon \to 0$, the spectrum of the Jacobian will also be positive definite indicating the point $v_i^\perp$ is a repeller. This is indicated by the blue arrows in Figure 7b.

Figure 7b summarizes the results that the stable region for $\alpha$-EigenGame consists of an $\mathcal{O}(\epsilon)$ ball around the true optimum for parents with $\mathcal{O}(\epsilon)$ angular error. $\qquad\square$

# E  Noise is Martingale Difference Sequence

Let $\Delta_i^{\mu,t} = \left[I - \sum_{j<i} \hat{v}_j^{(t)} \hat{v}_j^{(t)\top}\right] C \hat{v}_i^{(t)}$ be the $\mu$-EigenGame update direction computed using the full expected covariance matrix. Let $\hat{\Delta}_i^{\mu,t} = \left[I - \sum_{j<i} \hat{v}_j^{(t)} \hat{v}_j^{(t)\top}\right] C_t \hat{v}_i^{(t)}$ be the update direction computed using a minibatch estimate of the covariance matrix where minibatches are unbiased because they are formed from data sampled uniformly at random from the dataset. Define $M_{i,t+1}^{V^{(t)}} = \hat{\Delta}_i^{\mu,t} - \Delta_i^{\mu,t}$ and let $M_{t+1}^{V^{(t)}} = [M_{1,t+1}^{V^{(t)}}, \ldots, M_{k,t+1}^{V^{(t)}}]^\top$ where $V^{(t)} = \{v_{i\in[k]}^{(t)}\}$.

**Lemma 4.** $\{M_{t+1}^{V^{(t)}}\}$ *is a bounded martingale difference sequence with respect to the increasing $\sigma$-fields*

$$\mathcal{F}_t = \sigma(\{(\hat{v}_i)_{i\in[k]}^{(0)}, \ldots, (\hat{v}_i)_{i\in[k]}^{(t)}\}, \{\hat{C}^{(1)}, \ldots, \hat{C}^{(t)}\}) \tag{28}$$

*Proof.* Given the filtration $\mathcal{F}_t$, we find

$$\mathbb{E}[M_{i,t+1}^{V^{(t)}}|\mathcal{F}_t] = \left[I - \sum_{j<i} \hat{v}_j^{(t)} \hat{v}_j^{(t)\top}\right] \mathbb{E}[C_t - C]\hat{v}_i^{(t)} = 0 \tag{29}$$

where the first equality holds because each $C_t$ is formed from a minibatch sampled i.i.d. from the dataset and therefore independent of the filtration. This result holds for all $i \in [k]$, therefore

$$\mathbb{E}[M_{t+1}^{V^{(t)}}|\mathcal{F}_t] = 0. \tag{30}$$

Furthermore,

$$\sup_t \mathbb{E}[||M_{i,t+1}^{V^{(t)}}||^2|\mathcal{F}_t] \tag{31}$$

$$= \sup_t \mathbb{E}\left[\hat{v}_i^{(t)\top}(C_t - C)^\top \overbrace{\left[I - \sum_{j<i} \hat{v}_j^{(t)} \hat{v}_j^{(t)\top}\right]^\top}^{P^\top} \overbrace{\left[I - \sum_{j<i} \hat{v}_j^{(t)} \hat{v}_j^{(t)\top}\right]}^{P}(C_t - C)\hat{v}_i^{(t)} \Big| \mathcal{F}_t\right] \tag{32}$$

$$= \sup_t \mathbb{E}[\hat{v}_i^{(t)\top}(C_t - C)^\top P^\top P(C_t - C)\hat{v}_i^{(t)}|\mathcal{F}_t] \tag{33}$$

$$\leq \sup_t \mathbb{E}[\lambda_i^2 \hat{v}_i^{(t)\top}(C_t - C)^\top I(C_t - C)\hat{v}_i^{(t)}|\mathcal{F}_t] \tag{34}$$

$$\leq \sup_t \lambda_i^2 \hat{v}_i^{(t)\top} \mathbb{E}[(C_t - C)^\top(C_t - C)|\mathcal{F}_t]\hat{v}_i^{(t)} \tag{35}$$

$$\leq \max\{1, (i-2)^2\}^2 \xi^2 \tag{36}$$

where $\lambda_i \leq \max\{1, (i-2)^2\}$ is the max singular value of $P^1$ and $\xi^2$ is the maximum eigenvalue of $\mathbb{E}[(C_t - C)^\top(C_t - C)]$ over all $t$. Summing over $i$ we find

$$\sup_t \mathbb{E}[||M_{i,t+1}^{V^{(t)}}||^2|\mathcal{F}_t] \leq (\sum_i \lambda_i)\xi^2 \tag{37}$$

$$\leq (2 + \sum_{a=1}^{k-2} a^2)\xi^2 \tag{38}$$

$$= (2 + \frac{1}{6}(k-2)(k-1)(2k-3))\xi^2 \tag{39}$$

$$\leq (2 + \frac{1}{3}k^3)\xi^2. \tag{40}$$

$\square$

---

[1] In the worst case, each subspace subtracted off by $\hat{v}_j \hat{v}_j^\top$ subtracts a 1 from the eigenvalue of 1 of the identity matrix.

This is a worst case bound. As the parent eigenvectors converge, the max singular value of $P$ converges to 1 so that the variance of the magnitude of the martingale difference is upper bounded by $k\xi^2$.

*Note*: For a finite dataset of $n$ samples or for a distribution with bounded moments like the Gaussian distribution, $\xi$ will be finite. However, for other distributions like the Cauchy distribution, $\xi$ may be unbounded, so care should be taken when running $\mu$-EigenGame in different stochastic settings.

## F  JAX PSEUDOCODE

For the sake of reproducibility we have included pseudocode in Jax. We use the Optax[2] optimization library Hessel *et al.* (2020) and the Jaxline training framework[3]. Our graph algorithm is a straightforward modification of the provided pseudo-code. See section G for details.

```
1  """
2  Copyright 2020 The EigenGame Unloaded Authors.
3
4
5  Licensed under the Apache License, Version 2.0 (the "License");
6  you may not use this file except in compliance with the License.
7  You may obtain a copy of the License at
8
9  https://www.apache.org/licenses/LICENSE-2.0
10
11 Unless required by applicable law or agreed to in writing, software
12 distributed under the License is distributed on an "AS IS" BASIS,
13 WITHOUT WARRANTIES OR CONDITIONS OF ANY KIND, either express or implied.
14 See the License for the specific language governing permissions and
15 limitations under the License.
16 """
17
18 import jax
19 import optax
20 import jax.numpy as jnp
21
22 def eg_grads(vi: jnp.ndarray,
23               weights: jnp.ndarray,
24               eigs: jnp.ndarray,
25               data: jnp.ndarray) -> jnp.ndarray:
26     """
27     Args:
28      vi: shape (d,), eigenvector to be updated
29      weights:  shape (k,), mask for penalty coefficients,
30      eigs: shape (k, d), i.e., vectors on rows
31      data: shape (N, d), minibatch X_t
32     Returns:
33      grads: shape (d,), gradient for vi
34     """
35   weights_ij = (jnp.sign(weights + 0.5) - 1.) / 2.  # maps -1 to -1 else
        to 0
36   data_vi = jnp.dot(data, vi)
37   data_eigs = jnp.transpose(jnp.dot(data,
38                             jnp.transpose(eigs)))  # Xvj on row j
39   vi_m_vj = jnp.dot(data_eigs, data_vi)
40   penalty_grads = vi_m_vj * jnp.transpose(eigs)
41   penalty_grads = jnp.dot(penalty_grads, weights_ij)
42   grads = jnp.dot(jnp.transpose(data), data_vi) + penalty_grads
43   return grads
44
45
46 def utility(vi, weights, eigs, data):
```

[2]https://github.com/deepmind/optax
[3]https://github.com/deepmind/jaxline

```
47      """Compute Eigengame utilities.
48      util: shape (1,), utility for vi
49      """
50    data_vi = jnp.dot(data, vi)
51    data_eigs = jnp.transpose(jnp.dot(data, jnp.transpose(eigs)))  # Xvj on
          row j
52    vi_m_vj2 = jnp.dot(data_eigs, data_vi)**2.
53    vj_m_vj = jnp.sum(data_eigs * data_eigs, axis=1)
54    r_ij = vi_m_vj2 / vj_m_vj
55    util = jnp.dot(jnp.array(r_ij), weights)
56    return util
```

Listing 1: Gradient and utility functions.

```
1  def _grads_and_update(vi, weights, eigs, input, opt_state,
       axis_index_groups):
2      """Compute utilities and update directions, psum and apply.
3      Args:
4       vi: shape (d,), eigenvector to be updated
5       weights:  shape (k_per_device, k,), mask for penalty coefficients,
6       eigs: shape (k, d), i.e., vectors on rows
7       input: shape (N, d), minibatch X_t
8       opt_state: optax state
9       axis_index_groups: For multi-host parallelism https://jax.
       readthedocs.io/en/latest/_modules/jax/_src/lax/parallel.html
10      Returns:
11       vi_new: shape (d,), eigenvector to be updated
12       opt_state: new optax state
13       utilities: shape (1,), utilities
14      """
15      grads, utilities = _grads_and_utils(vi, weights, V, input)
16      avg_grads = jax.lax.psum(
17          grads, axis_name='i', axis_index_groups=axis_index_groups)
18      vi_new, opt_state, lr = _update_with_grads(vi, avg_grads, opt_state)
19      return vi_new, opt_state, utilities
20
21  def _grads_and_utils(vi, weights, V, inputs):
22      """Compute utiltiies and update directions ("grads").
23          Wrap in jax.vmap for k_per_device dimension."""
24      utilities = utility(vi, weights, V, inputs)
25      grads = eg_grads(vi, weights, V, inputs)
26      return grads, utilities
27
28  def _update_with_grads(vi, grads, opt_state):
29      """Compute and apply updates with optax optimizer.
30          Wrap in jax.vmap for k_per_device dimension."""
31      updates, opt_state = self._optimizer.update(-grads, opt_state)
32      vi_new = optax.apply_updates(vi, updates)
33      vi_new /= jnp.linalg.norm(vi_new)
34      return vi_new, opt_state
```

Listing 2: EigenGame Update functions.

```
1  def init(self, *):
2      """Initialization function for a Jaxline experiment."""
3      weights = np.eye(self._total_k) * 2 - np.ones((self._total_k, self.
       _total_k))
4      weights[np.triu_indices(self._total_k, 1)] = 0.
5      self._weights = jnp.reshape(weights, [self._num_devices,
6                                            self._k_per_device,
7                                            self._total_k])
8
9      local_rng = jax.random.fold_in(jax.random.PRNGkey(seed), jax.host_id
       ())
10      keys = jax.random.split(local_rng, self._num_devices)
```

```
11    V = jax.pmap(lambda key: jax.random.normal(key, (self._k_per_device,
      self._dims))))(keys)
12    self._V = jax.pmap(lambda V: V / jnp.linalg.norm(V, axis=1, keepdims=
      True))(V)
13
14    # Define parallel update function. If k_per_device is not None, wrap
      individual functions with vmap here.
15    self._partial_grad_update = functools.partial(
16        self._grads_and_update, axis_groups=self._axis_index_groups)
17    self._par_grad_update = jax.pmap(
18        self._partial_grad_update, in_axes=(0, 0, None, 0, 0, 0),
      axis_name='i')
19
20    self._optimizer = optax.sgd(learning_rate=1e-4, momentum=0.9,
      nesterov=True)
21
22 def step(self, *):
23    """Step function for a Jaxline experiment"""
24    inputs = next(input_data_iterator)
25    self._local_V = jnp.reshape(self._V, (self._total_k, self._dims))
26    self._V, self._opt_state, utilities, lr = self._par_grad_update(
27        self._V, self._weights_jnp, self._local_V, inputs, self.
      _opt_state,
28        global_step)
```

Listing 3: Skeleton for Jaxline experiment.

## G   $\mu$-EIGENGAME ON GRAPHS

Algorithm 3 receives a stream of edges represented as a matrix with edges on the rows and outgoing node id ($out$) and incoming node id ($in$) as nonegative integers on the columns. The method zeros_like($z$) returns an array of zeros with the same dimensions as $z$. The method index_add($z, idx, val$) adds the values in array $val$ to $z$ at the corresponding indices in array $idx$ with threadsafe locking so that indices in $idx$ may be duplicated. Both methods are available in JAX. The largest eigenvector $\hat{v}_1$ is learned to estimate $\lambda_1$ and may be discarded. The bottom-$k$ eigenvectors are returned by the algorithm in increasing order. Algorithm 3 expects $k + 1$ random unit vectors as input rather than $k$ in order to additionally estimate the top eigenvector necessary for the computation; otherwise, the inputs are the same as Algorithm 1.

## H   ALGORITHM DESIGN PROCESS

In section 4.1, we presented $u_i^\mu$ as the Rayleigh quotient of a deflated matrix (repeated in equation (8) for convencience):

$$u_i^\mu = \hat{v}_i^\top \overbrace{\Big[ I - \sum_{j<i} \hat{v}_j \hat{v}_j^\top \Big]}^{\text{deflation}} C \bullet [\hat{v}_i] \tag{41}$$

$$= \hat{v}_i^\top C \bullet [\hat{v}_i] - \sum_{j<i} (\hat{v}_i^\top C \hat{v}_j)(\bullet[\hat{v}_i]^\top \hat{v}_j) \tag{42}$$

$$u_i^\alpha = \overbrace{\hat{v}_i^\top C \hat{v}_i}^{\text{Var}} - \sum_{j<i} \overbrace{\frac{\langle \hat{v}_i, C \hat{v}_j \rangle^2}{\langle \hat{v}_j, C \hat{v}_j \rangle}}^{\perp\text{-penalty}} . \tag{43}$$

Alternatively, we can consider $u_i^\mu$ as equation (42) in light of the derivation for $u_i^\alpha$ by Gemp *et al.* (2021). In that case, utilities are constructed from entries in the matrix

---

**Algorithm 3** $\mu$-EigenGame for Graphs (w/o Riemannian gradient projection)

---

1: Given: Edge stream $\mathcal{E}_t \in \mathbb{R}^{n' \times 2}$, number of parallel machines $M$ per player (minibatch size per partition $n'' = \frac{n'}{M}$), initial vectors $\hat{v}_i^0 \in \mathcal{S}^{d-1}$, step size sequence $\eta_t$, and iterations $T$.
2: $\hat{v}_i \leftarrow \hat{v}_i^0$ for all $i \in \{1, \ldots, k+1\}$
3: $\lambda_1 \leftarrow 2|\mathcal{V}|$ *upper bound on top eigenvalue*
4: **for** $t = 1 : T$ **do**
5:     **parfor** $i = 1 : k+1$ **do**
6:         **parfor** $m = 1 : M$ **do**
7:             $[Xv]_i = \hat{v}_i(out_{tm}) - \hat{v}_i(in_{tm})$
8:             $[X^\top Xv]_i \leftarrow$ `zeros_like`$(\hat{v}_i)$
9:             $[X^\top Xv]_i \leftarrow$ `index_add`$([X^\top Xv], out_{tm}, [Xv]_i)$
10:           $[X^\top Xv]_i \leftarrow$ `index_add`$([X^\top Xv], in_{tm}, -[Xv]_i)$
11:           **if** $i = 1$ **then**
12:             $\lambda_1 \leftarrow ||[Xv]_i||^2$
13:             $\tilde{\nabla}_{it'}^\mu \leftarrow [X^\top Xv]_i$
14:           **else**
15:             $\tilde{\nabla}_{im}^\mu \leftarrow \lambda_1[\hat{v}_i - \sum_{1 < j < i}(\hat{v}_i^\top \hat{v}_j)\hat{v}_j]$
16:             $[Xv]_j = \hat{v}_j(out_{tm}) - \hat{v}_j(in_{tm})$ for all $j$
17:             $\tilde{\nabla}_{it'}^\mu -= [X^\top Xv]_i - \sum_{1 < j < i}([Xv]_i^\top [Xv]_j)\hat{v}_j$
18:           **end if**
19:         **end parfor**
20:         $\tilde{\nabla}_i^\mu \leftarrow \frac{1}{n'} \sum_{t'} [\tilde{\nabla}_{it'}^\mu]$
21:         $\hat{v}_i' \leftarrow \hat{v}_i + \eta_t \tilde{\nabla}_i^\mu$
22:         $\hat{v}_i \leftarrow \frac{\hat{v}_i'}{||\hat{v}_i'||}$
23:     **end parfor**
24: **end for**
25: return $\{\hat{v}_i | i \in \{2, \ldots, k+1\}\}$

---

$$\hat{V}^\top C\hat{V} = \begin{bmatrix} \langle \hat{v}_1, C\hat{v}_1 \rangle & \langle \hat{v}_1, C\hat{v}_2 \rangle & \ldots & \langle \hat{v}_1, C\hat{v}_d \rangle \\ \langle \hat{v}_2, C\hat{v}_1 \rangle & \langle \hat{v}_2, C\hat{v}_2 \rangle & \ldots & \langle \hat{v}_2, C\hat{v}_d \rangle \\ \vdots & \vdots & \ddots & \vdots \\ \langle \hat{v}_d, C\hat{v}_1 \rangle & \langle \hat{v}_d, C\hat{v}_2 \rangle & \ldots & \langle \hat{v}_d, C\hat{v}_d \rangle \end{bmatrix}. \tag{44}$$

It is argued that if $\hat{V}$ diagonalizes $M$ and captures maximum variance, then the diagonal $\langle \hat{v}_i, C\hat{v}_i \rangle$ terms must be maximized and the off-diagonal $\langle \hat{v}_i, C\hat{v}_j \rangle$ terms must be zero. As the latter mixed terms may be negative, the authors square the mixed terms to form "minimizable utilities" and divide them by $\langle \hat{v}_j, C\hat{v}_j \rangle$ so that they have similar "units" to the terms $\langle \hat{v}_i, C\hat{v}_i \rangle$ of the first type. In contrast, the $u_i^\mu$ utilities *could* be arrived at by instead multiplying the mixed terms by $\langle \hat{v}_i, \hat{v}_j \rangle$. While this ensures the mixed terms are positive with exact parents (because $\langle \hat{v}_i, C v_j \rangle = \lambda_j \langle \hat{v}_i, v_j \rangle$), it does not ensure they are always positive in general[4]. In other words, $u_i^\mu$ is defined in way such that the $\perp$-penalties actually encourage vectors to align at times when they should in fact do the opposite! We therefore consider it unlikely that anyone would pose equation (42) as a utility if coming from the perspective of $\alpha$-EigenGame.

We could have extended the diagram in Figure 5b to include this dead end link. We have also included the true gradient of $u_i^\mu$ as a logical endpoint. We present these extensions in Figure 8.

---

[4]e.g., let $C = \begin{bmatrix} 2 & 1 \\ 1 & 1 \end{bmatrix}$ and place $\hat{v}_1$ at $-30°$ and $\hat{v}_2$ at $90°$.

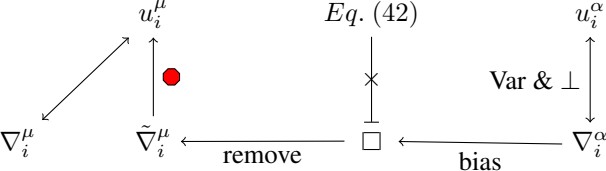

Figure 8: This diagram presents the relationships between utilities and updates. An arrow indicates the endpoint is reasonably derived from the origin; the lack of an arrow indicates the direction is unlikely. The link from equation (42) is explicitly crossed out with a hard stop for emphasis.

## H.1 GRADIENT ASCENT ON $u_i^\mu$

If we remove the stop gradient 🔴 from equation (41), we are left with equation (45):

$$u_i^\mu = \hat{v}_i^\top [\overbrace{I - \sum_{j<i} \hat{v}_j \hat{v}_j^\top}^{\text{deflation}}] C \hat{v}_i. \tag{45}$$

If we then differentiate this utility, we find its gradient is

$$\nabla_i^\mu = C\hat{v}_i - \frac{1}{2} \sum_{j<i} [(\hat{v}_i^\top C \hat{v}_j)\hat{v}_j + (\hat{v}_i^\top \hat{v}_j) C \hat{v}_j]. \tag{46}$$

We also reran experiments with this update direction, $\nabla_i^\mu$ on the synthetic and MNIST domains. The update is unbiased, so it would be expected to scale well, however, it (in orange) appears to scale more poorly than $\mu$-EigenGame with smaller minibatches.

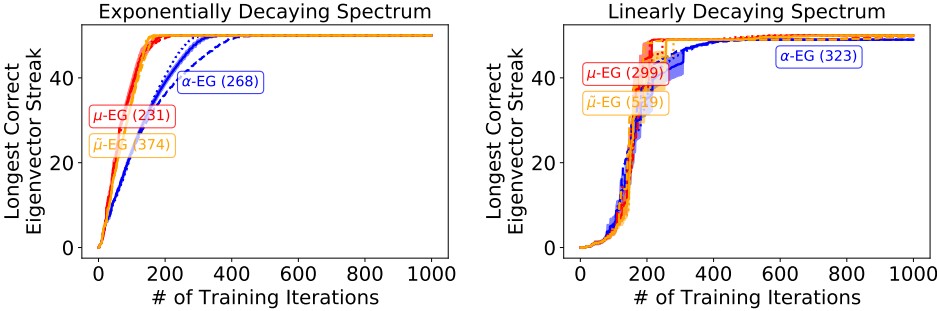

Figure 9: Synthetic Experiment. Runtime (milliseconds) in legend.

In Figure 10, $\tilde{\mu}$-EG appears to converge in terms of subspace error but slows in terms of longest eigenvector streak. $\tilde{\mu}$-EG updates are also unbiased so we would expect it is convergent globally, but it underperforms relative to $\mu$-EG. In contrast, $\alpha$-EG stalls in terms of subspace error likely due to bias.

Note that with exact parents, mu-EG and mu-tilde-EG have the same update (plug $Cvj = \lambda_j vj$ into equation (46)), so the difference must come from when the parents are still inaccurate.

In Figure 11, we have plotted the norm of the difference between subsequent values of the eigenvectors over training, i.e., how "far" $v_i$ moves after every update. Note all algorithms were run with the same fixed step size of $10^{-3}$, which was optimal for each algorithm in this setting. Clearly, the gradient version of $\alpha$-EigenGame ($\tilde{\mu}$-EG) shown in Figure 11c exhibits higher norms overall.

We believe this is due to the higher variance penalty terms (all methods maximize the same Rayleigh quotient term). Note that both $\alpha$-EG and $\mu$-EG construct their pentalty directions by a weighted sum of terms. These terms are computed differently, but both compute weights with inner products between $v_i$ and $v_j$ after projecting onto the samples in the minibatch $X_t$. For example, $\mu$-EG

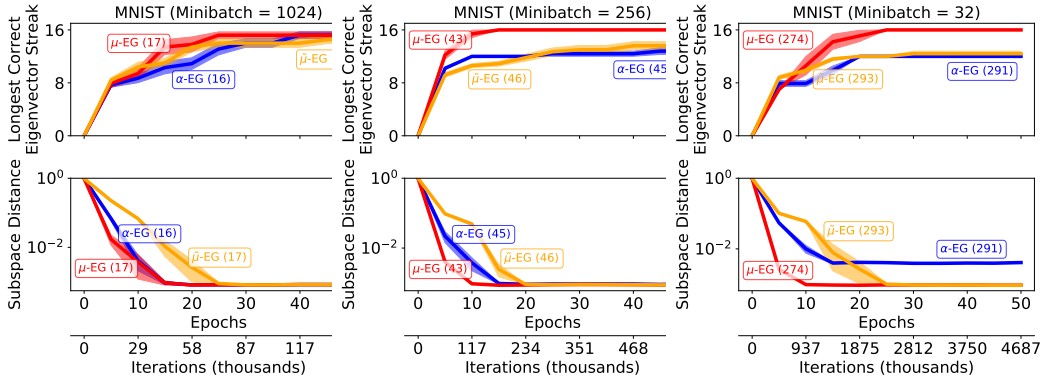

Figure 10: MNIST Experiment. Runtime (seconds) in legend. Each column evaluates a different minibatch size $\in \{1024, 256, 32\}$.

computes $\langle X_t v_i, X_t v_j \rangle = v_i^\top C_t v_j$. Without loss of generality, assume $C$ is a diagonal matrix (with the eigenvalues on its diagonal). Then $\langle v_i^\top C v_j \rangle = \sum_k \lambda_k v_{ik} v_{jk}$. The eigenvectors $v_i = e_i$ in this case, and so the inner product measures alignment between $v_i$ and $v_j$ in the dimensions that $v_i$ and $v_j$ are trained to be orthogonal. Due to noise in the minibatches $X_t$, $v_i$ and $v_j$ may "drift" in the remaining dimensions. Projecting essentially ignores these though because they are weighted by small eigenvalues.

In contrast, $\tilde{\mu}$-EG computes weights as raw inner products between $v_i$ and $v_j$. Therefore, any drift of $v_i$ and $v_j$ due to noise in the minibatch samples contributes to the inner product: $\langle v_i, v_j \rangle = \sum_k v_{ik} v_{jk}$. We suspect this is the reason $\tilde{\mu}$-EG exhibits higher drift distance.

In summary, $\alpha$-EG updates are computed as a weighted sum of terms where the weights are computed using inner products between $v_i$ and its parents after projecting them to a lower dimensional space. Computing the inner product in this particular space results in lower variance for each inner product. Unfortunately, $\alpha$-EG updates are biased, so while they exhibit relatively low "norm of drift", they converge to the incorrect solution (parents never converge to precise solution which prohibits children from learning accurately solutions).

$\tilde{\mu}$-EG updates are unbiased, so they should converge to the correct solution in the limit, but they exhibit higher variance due to their penalty weights being computed in the original high dimensional space (i.e., they pick up every little bit of noise).

Finally, $\mu$-EG updates are unbiased and compute their penalty weights in a lower dimensional space, suppressing the bulk of the noise that appears from drift caused by randomness in the minibatches. They exhibit the lowest levels of "norm of drift".

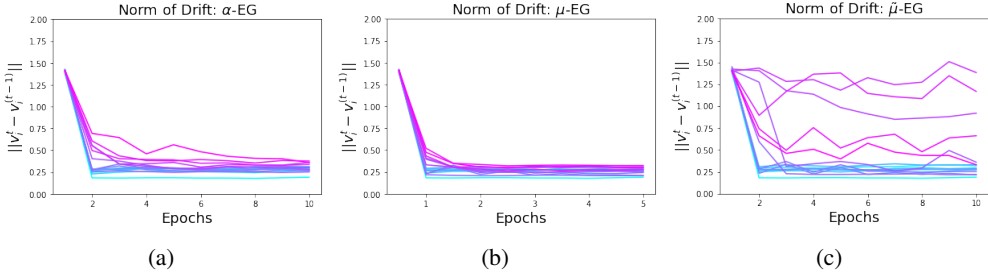

(a)  (b)  (c)

Figure 11: MNIST Experiment. Subfigures (a-c) correspond to $\mu$-EigenGame, $\alpha$-EigenGame, and the gradient version of $\alpha$-EigenGame discussed above respectively. Each experiment is conducted with a minibatch size of 32 over five epochs of training and averaged over 10 trials. Each curve shows the update distance after each iteration for one of the top-16 eigenvectors. Cyan curves indicate eigenvectors higher in the hierarchy (e.g., $v_1$) and magenta curves indicate eigenvectors lower in the hierarchy.

## H.2 ACCELERATION

We conjecture that $\mu$-EigenGame converges more quickly than $\alpha$-EigenGame because of the following two claims.

**Claim 1** The penalty terms of $\tilde{\nabla}_i^\mu$ are all within $90°$ of those of $\nabla_i^\alpha$ because $\left\langle \frac{C\hat{v}_j}{\hat{v}_j^\top C\hat{v}_j}, \hat{v}_j \right\rangle = 1 > 0$.

**Claim 2** The penalty terms of $\tilde{\nabla}_i^\mu$ are all smaller in magnitude than those of $\nabla_i^\alpha$: $||\hat{v}_j|| \leq \left|\left| \frac{C\hat{v}_j}{\hat{v}_j^\top C\hat{v}_j} \right|\right|$.

Indeed, consider the direction $C\hat{v}_j$. By properties of the vector rejection, we know the rejection of this direction onto the tangent space of the unit sphere has magnitude less than or equal to that of the original vector, $||C\hat{v}_j||$. The projection is $(I - \hat{v}_j\hat{v}_j^\top)(C\hat{v}_j)$. Therefore, the rejection is $\hat{v}_j\hat{v}_j^\top(C\hat{v}_j)$ and, by the preceding argument, we know its magnitude $|\hat{v}_j^\top C\hat{v}_j|||\hat{v}_j||$ is less than or equal to $||C\hat{v}_j||$. Rearranging the inequality completes the proof. $\square$

By **Claim 1**, the penalty directions of $\mu$-EG and $\alpha$-EG approximately agree. And by **Claim 2**, $\alpha$-EG's penalty direction is shorter. Consider a scenario where a parent of $\hat{v}_i$ has not converged and transiently occupies space along $\hat{v}_i$'s geodesic to its true endpoint $\hat{v}_i$, a strong penalty term will force $\hat{v}_i$ to take a roundabout trajectory, thereby slowing its convergence. A weaker penalty term allows $\hat{v}_i$ to pass through regions occupied by its parent **as long as** its parent is not an eigenvector. Recall from Section 3 that the two utilities are equivalent when the parents are eigenvectors.

