# OpenReview forum: "EigenGame Unloaded: When playing games is better than optimizing"
_ICLR.cc/2022/Conference — ICLR 2022 Poster_

### Official Review · Reviewer_JMvM · 2021-10-24

**Correctness:** 4
**Technical Novelty And Significance:** 3
**Empirical Novelty And Significance:** 3
**Recommendation:** 8
**Confidence:** 4

**Main Review:**

Regarding strengths, the authors clearly explain the weakness of the existing $\alpha$- EigenGame algorithm and update. They also elucidate the core intuition of why their simple yet effective solution works (Lemma 1) and how their approach can be interpreted as a deflation technique commonly used by traditional algorithms for SVD (Equation 7). While these intuitions are probably more than sufficient to deduce that $\mu$-EigenGame converges in practice, establishing the same results theoretically requires non trivial additional work as showcased in sections C, D and E of the Appendix.

Regarding weaknesses, I agree with the statement of the authors in Appendix H.1 that further research is needed to understand what properties affect convergence rates in practice. It is clear that update bias and data parallelism do not solely determine the empirical convergence rate. Thus I think it would be beneficial to explore more detailed comparisons between the $\mu$-EigenGame, its variant $\tilde{\mu}$ and of course $\alpha$-EigenGame. For example, maybe the variance of the updates is the core contributor for the low performance of both $\alpha$-EigenGame and the $\tilde{\mu}$-EigenGame for small batch sizes. While this comparison does not change the fact that $\mu$-EigenGame outperforms the alternatives, it would be beneficial for the community to understand the practical reasons in more detail.

I have read the author's response. I find that the updated version of Appendix H improves the paper and I would encourage to incorporate elements of it or at least some of the takeaways in the main paper. I remain in favor of acceptance.


**Summary Of The Paper:**

The authors study the problem of finding the top $k$ right singular vectors of a data matrix $X$. They propose a modification of an established game theoretic gradient based algorithm $\alpha$-EigenGame. They observe that the gradients of $\alpha$-EigenGame are biased when implemented stochastically by subsampling the data matrix $X$. Their proposed modification guarantees unbiased updates while converging to the true singular vectors. Their modified updates in combination with their data parallel distrusted algorithm leads to significantly improved convergence rates.

**Summary Of The Review:**

The authors address the problem of the biased updates of the $\alpha$-EigenGame in a simple yet effective and theoretically sound way. The proposed changes make the algorithm both highly data parallel and also very robust when using small batch sizes. While more work is definitely required to understand the underlying factors that affect convergence rates, I am confident that this work makes an important first step towards this goal.

---

> ### Author Response · Authors · 2021-11-16
> **Response**
>
> Thank you for your thought provoking review and for taking the time to inspect the appendix.
>
> *$\tilde{\mu}$-EG (gradient) vs $\mu$-EG (not gradient)*: TL;DR We have conducted additional analysis and believe $\tilde{\mu}$-EG performs worse due to higher variance penalty terms.
>
> We agree that the comparison in Appendix H is quite an interesting and counterintuitive result. Note that Figure 9 compares the three methods in two different full batch (non-stochastic settings). In those cases, $\mu$-EG and $\tilde{\mu}$-EG appear to perform very similarly. In the stochastic setting (mb size 32) in Figure 10, the true gradient method, $\tilde{\mu}$-EG, appears to converge in terms of subspace error but slows in terms of longest eigenvector streak. $\tilde{\mu}$-EG is also unbiased so we would expect it is convergent globally, but it underperforms relative to $\mu$-EG. In contrast, $\alpha$-EG stalls in terms of subspace error likely due to bias. We have added the following discussion and plots to Appendix H.
>
> Note that with exact parents, $\mu$-EG and $\tilde{\mu}$-EG have the same update (plug $Cvj = \lambda_j vj$ into equation 46), so the difference must come from when the parents are still inaccurate.
>
> In Figure 11 (new), we have plotted the norm of the difference between subsequent values of the eigenvectors over training for the mb=32 setting averaged over 10 trials, i.e., how “far” $v_i$ moves after every update. Note all algorithms were run with the same fixed step size (1e-3 gave the highest average performance over 10 trials). The cyan curves correspond to eigenvectors higher in the hierarchy (e.g., $v_1$) and the magenta curves correspond to those lower in the hierarchy (e.g., $v_{16}$). Clearly, $\tilde{\mu}$ exhibits higher norms.
>
> We believe this is due to higher variance penalty terms (all methods maximize the same Rayleigh quotient term so it cannot be the culprit). Note that both $\alpha$-EG and $\mu$-EG construct their penalty directions by a weighted sum of terms. These terms are computed differently, but both compute weights with inner products between $v_i$ and $v_j$ after projecting onto the samples in the minibatch $X_t$. For example, $\mu$-EG computes $\langle X_t v_i, X_t v_j \rangle = v_i^\top C_t v_j$. Without loss of generality, assume $C$ is a diagonal matrix (with the eigenvalues on its diagonal). Then $\langle v_i^\top C v_j \rangle = \sum_k \lambda_k v_{ik} v_{jk}$. The eigenvectors $v_i = e_i$ in this case, and so the inner product measures alignment between $v_i$ and $v_j$ primarily in the dimensions that $v_i$ and $v_j$ are trained to be orthogonal. Due to noise in the minibatches $X_t$, $v_i$ and $v_j$ may “drift” in the remaining dimensions. Projecting essentially ignores these though because they are weighted by small eigenvalues.
> In contrast, $\tilde{\mu}$-EG computes weights as raw inner products between $v_i$ and $v_j$. Therefore, any drift of $v_i$ and $v_j$ due to noise in the minibatch samples contributes to the inner product: $\langle v_i, v_j \rangle = \sum_k v_{ik} v_{jk}$. We suspect this is the reason $\tilde{\mu}$-EG exhibits higher “norm of drift”.
>
> In summary, $\alpha$-EG updates are computed as a weighted sum of terms where the weights are computed using inner products between $v_i$ and its parents after projecting them to a lower dimensional space. Computing the inner product in this particular space results in lower variance for each inner product. Unfortunately, $\alpha$-EG updates are biased, so while they exhibit relatively low “norm of drift”, they converge to the incorrect solution (parents never converge precisely to the true solutions which prohibits children from learning accurate solutions).
>
> $\tilde{\mu}$-EG updates are unbiased, so they should converge to the correct solution in the limit, but they exhibit higher variance due to their penalty weights being computed in the original high dimensional space (i.e., they pick up every little bit of noise).
>
> Finally, $\mu$-EG updates are unbiased and compute their penalty weights in a lower dimensional space, suppressing the bulk of the noise that appears from drift caused by randomness in the minibatches. They exhibit the lowest levels of “norm of drift”.

---

### Official Review · Reviewer_T9Hp · 2021-11-01

**Correctness:** 4
**Technical Novelty And Significance:** 2
**Empirical Novelty And Significance:** 3
**Recommendation:** 5
**Confidence:** 3

**Main Review:**

The paper improves $\alpha$-EigenGame by proposing a novel unbiased variant $\mu$-EigenGame. The proposed method adopts unbiased and parallelizable updates in the stochastic setting and is guaranteed by global convergence. The ideas of the paper are interesting and the writing is good. However, given the prior work $\alpha$-EigenGame, the (technical) novelty of this paper is not so much. Also, the theoretical part of this paper is on the weak side.

Theorem 1 requires that covariance matrix C is positive definite with distinct eigenvalues. I think this assumption is a bit strong since the algorithm only computes the top-k eigenvectors. Why it requires all eigenvalues of C to be distinct? Actually, $\alpha$-EigenGame only requires top-k eigenvalues are distinct (Theorem 2.1 of (Gemp et al., 2021)). In addition, methods such as Oja's algorithm only require an eigengap between the k and k+1 eigenvalues.

Theorem 1 only gives an asymptotic convergence. I think providing a finite-sample convergence can largely increase the theoretical contribution of this paper.

**Summary Of The Paper:**

The paper considers PCA problem from a game-theoretic view and propose a novel algorithm ($\mu$-EigenGame) with stochastic convergence guarantees. The proposed method introduces an unbiased update which allows greater parallelism over data. The empirical results show that $\mu$-EigenGame outperforms its predecessor $\alpha$-EigenGame.

**Summary Of The Review:**

Though the idea of this paper is interesting, the weakness of the theoretical analysis reduces my criterion on it. I think this paper is a bit below the bar.

---

> ### Author Response · Authors · 2021-11-15
> **Response**
>
> Thank you for reviewing our paper and for your attention to the technical details.
>
> *Distinct Eigenvalues*: Thank you for pointing this out. We were unnecessarily loose in our assumptions, and in fact, only require positive eigengaps for the first $k$ eigenvalues (where eigengap $i$ $=g_i = \lambda_i - \lambda_{i+1}$). This requires only small modifications to the text in the proof of Theorems 1 and 2, nothing technically different. This change is pain free because our proofs stand on the shoulders of the proofs in Gemp et al 2021 via Lemma 1 in this work. We have edited the paper accordingly.
>
> Other methods, such as Oja’s, only require an eigengap for the $k$-th value because they measure convergence in terms of subspace error rather than eigenvector error (which is what we and Gemp et al 2021 measure convergence by). Note that for an eigengap of zero, a unique eigenvector does not exist because any vector belonging to a certain subspace is valid. For example, consider the identity matrix (which has zero eigengap for all eigenvalues). Any orthonormal set of vectors suffices as an eigenbasis, so it has no unique eigenvector basis. Methods converging in terms of subspace error only require the $k$th eigengap be positive, because that’s all that’s required to ensure the top-$k$ subspace is unique (any orthonormal set of vectors spanning that subspace will be a valid solution though).
>
> *Asymptotic Convergence*: Please see our discussion above in the general comments to reviewers.

---

### Official Review · Reviewer_Tfdg · 2021-11-01

**Correctness:** 4
**Technical Novelty And Significance:** 3
**Empirical Novelty And Significance:** 3
**Recommendation:** 8
**Confidence:** 3

**Main Review:**

Strengths:
- the paper is very easy to read, and the challenges and main ideas to address them are very clear. Lemmas 1 and 2 are very simple yet provide good intuition on how to approach the problem.
- the paper improves on previous work by showing how to do an unbiased update for eigengame, that can be run in parallel. This gets around the challenge from previous work that the update does not decompose cleanly over data partitions, due to non-linearities in the covariance matrix.
- the author show that SVD is the \emph{unique} Nash of the game they consider and define. This means there is no equilibrium selection problem, and their approach is guaranteed to find SVD and not another equilibrium.
- The experiments seem to show that the authors approach does better and coverges faster than previous work when it comes to the longest correct eigenvalue streak. The approach does better than standard Eigengame and not any worse than other approaches in terms of distance between the subspace defined by the true eigenvalues/eigenvectors and the one that is recovered.

Weaknesses:
- The main weakness of the paper is that the results are convergence results, and do not deal with finite samples. It might be nice to write down some concentration bounds on how well the authors' approach performs in-sample (though the authors' experiments give some insights as to this)
- Can the authors say a bit more about how they find an equilibrium, and why do they expect their approach to find one? Computing an equilibrium of general, non 2-player zero-sum games is known to be hard (even if this is only true in the worst-case). Being able to compute best responses is generally a much easier problem than finding their intersection. I was wondering if there was some intuition behind how the authors find one.

**Summary Of The Paper:**

The paper studies "EigenGame", which is a game-theoretic approach to eigenvalue decomposition. This paper extends EigenGame to parallel settings; the authors propose an unbiased stochastic update rule to compute eigenvalues of a matrix in parallel, that outperforms previous approaches.

**Summary Of The Review:**

Overall I think this is a novel and interesting contribution, that solves an important problem and provides a framework that improves over previous work.

---

> ### Author Response · Authors · 2021-11-15
> **Response**
>
> Thank you for reviewing our paper and for your questions regarding where EigenGame sits within the context of computing Nash equilibria more generally.
>
> *Convergence Results*: Please see our discussion above in the general comments to reviewers.
>
> *Computing a Nash Equilibrium*: Correct, approximating a Nash equilibrium is difficult for a general game (it belongs to the complexity class PPAD), however, as you hint at, this is not necessarily true for specific game classes, for example, monotone games or polymatrix games for which an approximate equilibrium can be obtained in polynomial time [2,3]. In this work, we have constructed an EigenGame such that its Nash equilibrium can be approximated with vanishing error. This is primarily due to two factors which we briefly touch on in the paragraph below the proof of Theorem 2.
>
> *a) Imposed Hierarchy*-- One factor is that the hierarchy reduces each player’s problem of computing a best response to an unknown set of opponent strategies to a problem of computing a best response to a fixed set of opponent strategies, .e.g, the Nash equilibrium.  It may be easier to understand this with a visual (see for instance the tweet for the original EigenGame paper, or Figure 2 and 4 in said paper). Imagine 3 players are competing in a hunt for 3 treasures on a sphere (blue=+3, red=+2, green=+1), but receive zero reward if they pick the same treasure. The Nash equilibria are any permutation of the 3 players picking distinct treasures. However, by imposing a hierarchy and, e.g., removing player 1’s penalty if they choose the same treasure as another player, we effectively change the game so one of the permutations becomes the unique Nash. If player 1 never receives a penalty, blue is a dominant strategy and therefore, must be its strategy in the Nash equilibrium. Given that player 1 plays blue and player 2 is not penalized for aligning with player 3, player 2 is free to choose among red and green. Clearly, it chooses red as it is a dominant strategy leaving player 3 with green. This story mimics the induction proof in Theorem 2.
>
> *b) Best response is learnable with Gradient Ascent*-- The other factor enabling computing the Nash equilibrium is that the best response for each player can be learned by gradient ascent. This is possible because each player’s utility function (as proven in $\alpha$-EG) is a cosine with period pi (i.e., $\cos(2\theta)$). Although a cosine is non-convex, it has a unique maximum (up to $\pi$ in this case) which can be solved for with gradient ascent.
>
> - [2] “Convex Optimization, Game Theory, and Variational Inequality Theory” by Scutari et al 2010
> - [3] “Computing Approximate Nash Equilibria in Polymatrix Games” by Deligkas et al 2014

---

> > ### Comment · Reviewer_Tfdg · 2021-11-24
> > **Ackowledge author response**
> >
> > Thank you very much for providing more details on how to compute a Nash here + discussing/adding intuition for finite sample convergence! My opinion is still that the contribution of the paper is interesting and I believe it should be accepted.

---

### Official Review · Reviewer_ub8e · 2021-11-02

**Correctness:** 4
**Technical Novelty And Significance:** 3
**Empirical Novelty And Significance:** 3
**Recommendation:** 5
**Confidence:** 3

**Main Review:**

The paper is very well written. It has a concise introduction and clear motivations to the problem, the proposed solution, and the final results. I am not in this area and I didn’t read the EigenGame before but I can quickly get the main contribution and the key observation of the paper.

Although this paper makes the story of EigenGame more complete, I think it is arguable that the contributions meet the acceptance threshold. The introduction of the unbiased estimation (Lemma 2) is nice, but it seems not very surprising given the EigenGame results. I think it would be great if the authors can highlight some technical contributions in the proof of Theorem 1.  Another weakness is that the convergence is asymptotic. I think the authors may also briefly discuss the difficulty of getting non-asymptotic convergence.

Looking at Figure 2 (a), it seems like the vectors in the \alpha-EigenGame can be learned in parallel, but the remark below Theorem 1 says the deterministic result in Gemp et al. (2021) requires that each v is learned in sequence. I wonder which one is the case in their paper.

**Summary Of The Paper:**

The authors extend previous work EigenGame from full-batch updates to stochastic updates. They propose an unbiased stochastic update that is asymptotically equivalent to the deterministic update, but it allows better parallelism.

**Summary Of The Review:**

Although the paper is interesting and well written, I have some concerns regarding the theoretical contributions. Therefore, I think the paper is slightly below the bar.

---

> ### Author Response · Authors · 2021-11-15
> **Response**
>
> Thank you for taking the time to review our paper. We’re pleased to hear you were able to easily follow the paper without having read Gemp et al 2021 beforehand. Note that we have addressed some of your questions in the general comment to the reviewers above.
>
> *“Lemma 2 is nice, but unsurprising”* - Could you clarify your comment? Do you mean the improved performance of $\mu$-EG over $\alpha$-EG is unsurprising given Lemma 2? Or do you mean it’s unsurprising that a derivation of an unbiased version, $\mu$-EG, of $\alpha$-EG was possible? Or something else?
>
> *Contributions of Convergence Theorem and Difficulties Obtaining Finite Sample Rates*: As you point out, we briefly compare / contrast our convergence proof with $\alpha$-EG below Theorem 1, but we can expand on it in our revision as we believe this is a strength, not a weakness of our paper. Please see our discussion of convergence above in the general comments to reviewers.
>
> - [1] “Convergence Analysis of Riemannian Stochastic Approximation Schemes” by Durmus et al 2021

---

### Author Response · Authors · 2021-11-15
**General Comments to Reviewers**

Thank you all for your comments and constructive criticisms. Several of you shared clarifying questions on convergence of $\mu$-EigenGame, so we address those together below. For your more unique comments, we have added responses below your individual reviews.

*Contributions of Convergence Theorem*-- Note we have added the following discussion to Appendix C. While the convergence proof for $\alpha$-EG (Gemp et al 2021) provides finite-sample rates, it is only valid for when the algorithm is applied sequentially (not parallelized over eigenvectors) and in the deterministic setting (batch size = entire data set). The experiments in $\alpha$-EG apply the algorithm in parallel and with mini batch sizes, meaning the $\alpha$-EG theorem does not actually apply to their experimental setting. That is to say, the $\alpha$-EG paper proposes updating eigenvectors in parallel in practice despite the lack of convergence guarantee.

In contrast, our convergence theorem applies to $\mu$-EG when applied in parallel (over the eigenvectors) and in the stochastic setting (with mini batch sizes), which is what we examine empirically in our experiments. The downside, as you point out, is that we do not provide finite-sample convergence rates.

Although we do not provide convergence rates, Lemma 1 proves that the $\mu$-EG update converges to the $\alpha$-EG update for each eigenvector, so intuitively, we expect the convergence rates to be relatively similar given that the algorithms are equivalent in the limit. Note that in the full batch setting where stochasticity does not conflate the differences between the two algorithms, Figure 6 in Appendix A empirically supports the similarity of the convergence rates for the two algorithms. Figure 3a which looks at a large (but not full) minibatch size, also shows a small difference between convergence for the two algorithms.

In summary, the $\alpha$-EG convergence theorem is impractical -- it provides convergence rates for a (non-parallel) algorithm in the (non-stochastic) setting, which is a combination that the $\alpha$-EG paper does not suggest be applied in practice. In contrast, our $\mu$-EG convergence theorem is practical -- it provides asymptotic convergence for a parallel algorithm in the stochastic setting.

*Difficulties Obtaining Finite Sample Rates*-- For finite sample convergence, we have been in contact with the authors of [1]. The primary obstacle is the construction of a suitable Lyapunov function to satisfy their Assumption A.2 on p. 4. Constructing Lyapunov functions is typically a difficult, tedious process. The paper [1] is very recent and finite sample convergence of Riemannian stochastic approximation (i.e., update directions are not gradients of any function) schemes is cutting edge, highly technical research. This is in contrast to Riemannian optimization (i.e., update directions are gradient of a function), which is much more mature. We hope theory advances in the next few years to a point where we can more easily provide convergence rates for algorithms like $\mu$-EG. We have added this discussion to Appendix C.

---

### Decision · Program_Chairs · 2022-01-20

**Decision:**

Accept (Poster)

**Comment:**

Motivated by the recently proposed EigenGame, this paper proposes an unbiased stochastic update to replace the biased one in the original EigenGame. The new algorithm is asymptotically equivalent to EigenGame, enjoys better parallelism on big data, and beats EigenGame in experiments. Some reviewers are originally concerned about the lack of finite sample convergence results. After the author's response and reviewer discussion, this paper does get sufficient support. Therefore, I recommend acceptance and encourage the authors to think about how to deliver a finite-sample analysis in future work.